# GPLD1 is a scavenger carrier mediating lysosomal degradation of extracellular aberrant proteins

Mizuki Tsuchiya[1], Yoichiro Yagishita[2], Eisuke Itakura[3,4,5]

**Loss of proteostasis leads to the accumulation of aberrant proteins, including aggregated proteins and amyloid fibrils, contributing to various diseases. Protein quality control systems are essential for maintaining proteostasis. Although intracellular mechanisms are well characterized, pathways responsible for the degradation of aberrant proteins outside the cell remain poorly understood. We previously identified the chaperone/carrier- and receptor-mediated extracellular protein degradation pathway, in which the extracellular chaperone clusterin binds misfolded proteins and the resulting complex is delivered to lysosomes via endocytosis. However, it remains unclear whether other factors are involved in this pathway. To identify novel regulators, plasma factors binding serum amyloid A1 were investigated. Glycosylphosphatidylinositol-specific phospholipase D1 (GPLD1) was found to directly bind serum amyloid A1 and promote its lysosomal degradation. This activity was independent of GPLD1's cleavage activity for GPI-anchored proteins. Furthermore, GPLD1 mediates lysosomal degradation of misfolded proteins, with cell surface heparan sulfate acting as its receptor. Our data demonstrate that GPLD1 is a novel scavenger carrier with substrate specificity distinct from clusterin, responsible for degrading extracellular aberrant proteins.**

## Introduction

The accumulation of aberrant proteins leads to the formation of aggregates, which underlie the pathogenesis of various neurodegenerative diseases (Tyedmers et al, 2010; Hipp et al, 2019). For instance, amyloid fibrils, composed of β-sheet-rich proteins, deposit in tissues and cause a condition known as amyloidosis (Sawaya et al, 2021). To maintain proteostasis, cells rely on intracellular protein quality control systems, including autophagy and the ubiquitin-proteasome system, which degrade misfolded or damaged proteins (Wolff et al, 2014; Klionsky et al, 2021). The cytoplasm is densely populated with proteins, reaching concentrations of ~200–300 g/liter.

Similarly, extracellular compartments, such as the bloodstream, also exhibit high protein concentrations at ~80 g/liter (Ellis, 2001). Although intracellular protein quality control mechanisms are well-characterized (Rusilowicz-Jones et al, 2022), the pathways responsible for the clearance of aberrant proteins in the extracellular environment remain poorly understood (Wyatt et al, 2013). In our previous study, we identified a novel pathway termed chaperone/carrier-mediated extracellular protein degradation (CRED) (Itakura et al, 2020). In this pathway, the extracellular chaperone clusterin recognizes and binds misfolded proteins, forming a complex that is subsequently internalized and selectively degraded in lysosomes via the cell surface heparan sulfate (HS) receptor. Clusterin also facilitates lysosomal degradation of amyloid β, a prototypical amyloidogenic protein (Calero et al, 2000; Itakura et al, 2020). This mechanism contributes to the clearance of misfolded proteins from the extracellular environment (Itakura et al, 2020). We have also found that α2-macroglobulin facilitates lysosomal degradation of extracellular aberrant proteins (Tomihari et al, 2023). However, the involvement of additional proteostasis regulators in the CRED pathway remains incompletely understood. In the present study, we investigated the existence of a novel CRED factor responsible for the degradation of extracellular amyloidogenic proteins. Amyloidosis is a pathological condition characterized by the extracellular deposition of insoluble, fibrillar proteins (amyloid fibrils), which leads to progressive organ dysfunction (Louros et al, 2023). In humans, more than 30 proteins have been identified as precursors of amyloid fibrils (Sipe et al, 2016). Serum amyloid A (SAA) is one such protein, strongly associated with amyloidosis and often arising as a complication of chronic inflammatory diseases. These conditions are marked by the deposition of insoluble amyloid fibrils in affected organs and tissues (Sun & Ye, 2016). Amyloid A (AA) amyloidosis specifically results from prolonged inflammation, which drives sustained overproduction of SAA1. This overproduction ultimately leads to the aggregation and deposition of AA amyloid in tissues (Gaffney, 2017). Human *SAA1* is encoded by one of four *SAA* genes. Among these, *SAA1* and *SAA2* are acute-phase genes that are transcriptionally up-regulated in response to inflammatory cytokines, which are induced by infections or autoimmune diseases. These proteins play roles in modulating cytokine synthesis and neutrophil chemotaxis (Uhlar & Whitehead, 1999). Consequently,

---

[1]Department of Biology, Graduate School of Science and Engineering, Chiba University, Chiba, Japan   [2]Department of Biology, Faculty of Science, Chiba University, Chiba, Japan   [3]Department of Biology, Graduate School of Science, Chiba University, Chiba, Japan   [4]Center of Quantum Life Science for Structural Therapeutics (cQUEST), Chiba University, Chiba, Japan   [5]Institute for Advanced Academic Research (IAAR), Chiba University, Chiba, Japan

Correspondence: eitakura@chiba-u.jp

identifying mechanisms responsible for the degradation of SAA1 in the bloodstream is of critical importance for the prevention and mitigation of amyloidosis and related inflammatory conditions. Glycosylphosphatidylinositol-specific phospholipase D1 (GPLD1) is a secreted protein of ~110 kD, predominantly produced by the liver (LeBoeuf et al, 1998). It possesses a catalytic site that cleaves the lipid-binding domain of glycosylphosphatidylinositol (GPI)-anchored proteins, which are exposed on the extracellular surface of the cell membrane. This enzymatic activity facilitates the release of GPI-anchored proteins from the cell surface (Cao et al, 2023). Recent studies have shown that plasma levels of GPLD1 increase after physical exercise (Horowitz et al, 2020; Ren et al, 2024). Moreover, the exogenous expression of GPLD1 in aged mice has been reported to enhance cognitive function (Horowitz et al, 2020). In this study, we identified GPLD1 as a plasma factor that binds SAA1, using mass spectrometry-based analysis. By developing an internalization assay using fluorescently tagged GPLD1, we demonstrated that GPLD1 promotes lysosomal degradation of SAA1 and other misfolded proteins. Mechanistically, HS functions as a cell surface receptor that mediates internalization of the GPLD1–aberrant protein complex. Notably, GPLD1 exhibited greater internalization activity toward SAA1 than clusterin, suggesting that SAA1 is a physiological substrate of GPLD1. These findings indicate that GPLD1 possesses distinct substrate specificity compared with the known extracellular chaperone clusterin and functions as a novel scavenger carrier involved in the clearance of extracellular aberrant proteins.

# Results

### GPLD1 binds SAA1 in plasma proteins

To identify novel extracellular proteostasis regulators, we analyzed SAA1-binding proteins from bovine plasma, which is rich in extracellular proteins, using His-ALFA-tagged SAA1-N as bait; this bait consists of the N-terminal 76 aa residues, which represent the core amyloidogenic fragment of SAA1 (Westermark et al, 1992; Liberta et al, 2019). Given the amyloidogenic nature of SAA1-N, ALFA-SAA1-N was purified from *Escherichia coli* under denaturing conditions in the presence of urea. Note that SAA1-N was prepared in 8 M urea-denatured form for all experiments. We confirmed that subsequent incubation of this denatured SAA1-N in PBS successfully induced the formation of protein aggregates (Fig S1A). The purified ALFA-SAA1-N protein was then incubated with bovine plasma (the carry-over concentration of urea from the SAA1-N stock was kept below 0.05 M in the final incubation medium) and subjected to immunoprecipitation using ALFA nanobody-conjugated beads. Ruby staining revealed a range of proteins bound to ALFA-SAA1-N (Fig S1B), and these were subsequently identified by mass spectrometry (Table S1). Notably, clusterin and apolipoprotein E—both known to interact with extracellular aberrant proteins—were among the identified proteins (Matsubara et al, 1996; Mahley et al, 2009), suggesting that this dataset may contain previously unrecognized extracellular proteostasis regulators involved in proteostasis. After excluding well-characterized proteins (APOB, APMAP, PLA2G7, GSN, VTN, COL6A3, COL18A1, and

C9), we shortlisted seven candidate proteins (GPLD1, AHSG, APOD, SERPIND1, SERPINA10, SERPINF2, and SPP2) as potential extracellular proteostasis regulators. We then examined whether these candidates were internalized in cells in an SAA1-N-dependent manner and subsequently degraded in lysosomes. To detect lysosomal uptake of the candidates, we developed an internalization assay using tandem-tagged constructs comprising red fluorescent protein (RFP:mCherry), green fluorescent protein (GFP:sfGFP), and a His-tag. RFP is highly resistant to lysosomal proteases and acidic pH, whereas GFP is more susceptible to degradation. Therefore, upon internalization and trafficking of RFP-GFP-His-tagged candidates into lysosomes, only the RFP portion accumulates, whereas the GFP signal is lost (Fig 1A). The RFP-GFP-His (RG)-tagged candidate proteins were overexpressed in Flp-In T-REx HEK293 cells. The conditioned medium containing RG-tagged fusion proteins was collected and purified using His-tag affinity chromatography. For the internalization assay, ALFA-SAA1-N was mixed with serum-free medium containing the RG-tagged fusion protein and incubated at 37°C for 2 h before being added to cultured cells. After 18 h of incubation, the cells were harvested, and GFP and RFP fluorescence intensities were measured using flow cytometry. The internalization assay revealed that one candidate, GPLD1-RG, exhibited an SAA1-N-dependent increase in RFP fluorescence intensity (Fig 1B and C). Internalization of GPLD1-RG positively correlates with the dose of SAA1-N (Fig S1C). Given that the SAA1-N stock solution contains 8 M urea, resulting in a final assay concentration of less than 50 mM, we tested the impact of 50 mM urea on GPLD1-RG uptake. Control experiments showed that the presence of 50 mM urea did not alter GPLD1-RG internalization (Fig S2A), confirming that the residual urea does not interfere with the assay results. To investigate whether GPLD1-RG undergoes lysosomal degradation, cells were treated with bafilomycin A1 (BafA). BafA is a well-known inhibitor of lysosomal acidification that also affects endocytic trafficking (Baravalle et al, 2005). Consistent with its dual role in inhibiting both endocytosis and lysosomal function, BafA treatment led to a marked reduction in albumin internalization (Fig S2B). BafA treatment suppressed the SAA1-N-dependent increase in RFP accumulation, whereas GFP intensity increased (Fig 1C), suggesting that the lysosomal trafficking of GPLD1-RG is inhibited. We further verified the lysosomal degradation of GPLD1-RG biochemically using an RFP cleavage assay (Tomihari et al, 2021). This assay detects the release of free 25-kD RFP fragments generated from lysosomal digestion of the 160-kD GPLD1-RG fusion protein. Free RFP was detected in cells treated with GPLD1-RG (Fig 1D), and the amount of free RFP increased in the presence of SAA1-N. Notably, this increase was attenuated by BafA treatment. We investigated whether GPLD1 targets SAA1 during its denaturation process or after it has formed aggregates. To this end, GPLD1-RG was either mixed with ALFA-SAA1-N simultaneously or added to ALFA-SAA1-N that had been preincubated alone in the medium (Fig S2C). Internalization assays revealed that preincubated SAA1-N was internalized at levels comparable to those observed under the simultaneous mixing condition. These results suggest that GPLD1 is involved in the internalization of aggregated SAA1. These results indicate that the GPLD1–SAA1-N complex is internalized and subsequently degraded in lysosomes.

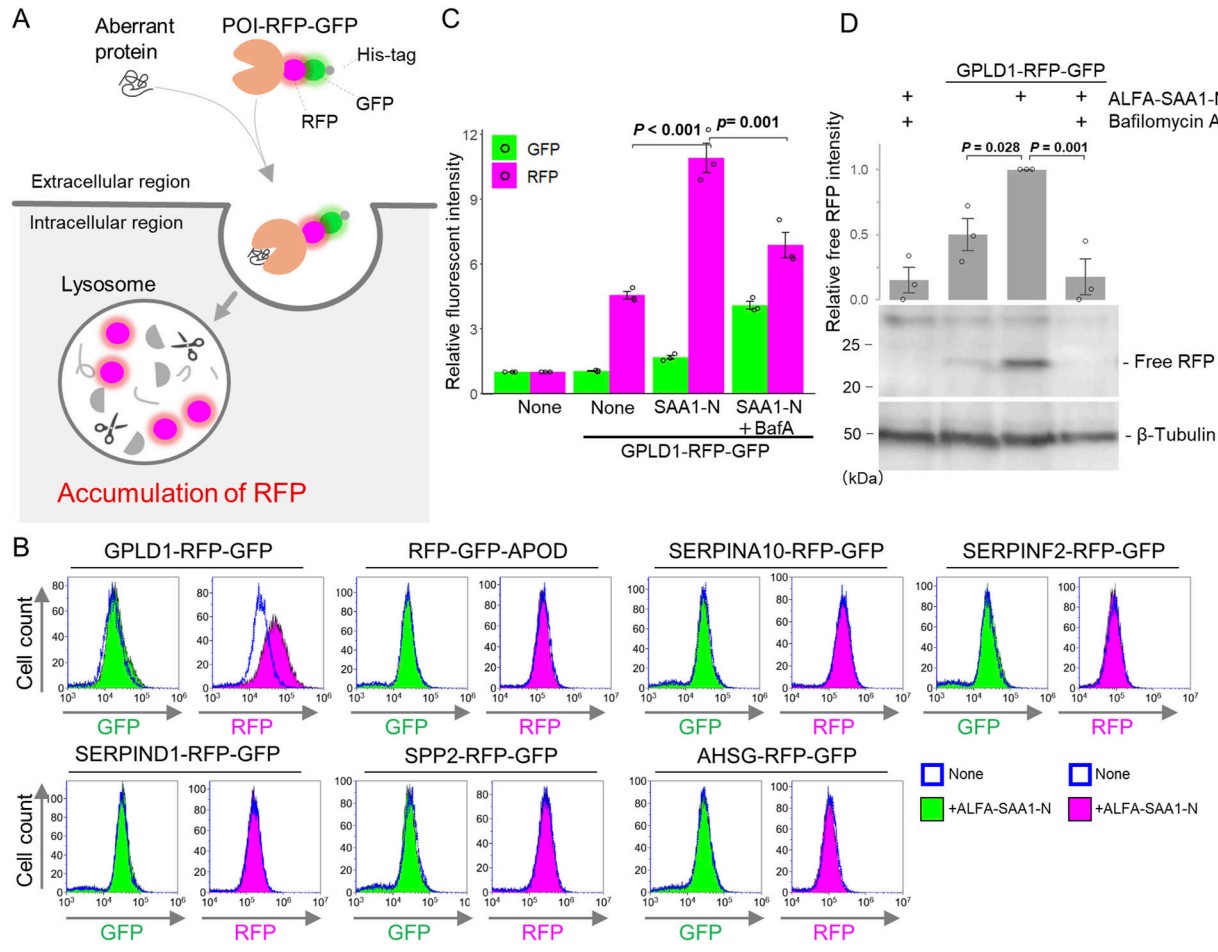

**Figure 1. Identification of GPLD1 as an extracellular scavenger carrier.**
**(A)** Schematic of the RFP-GFP internalization assay. Upon internalization, the protein of interest and GFP are degraded by lysosomal proteases, whereas RFP is resistant to lysosomal degradation and accumulates in lysosomes. **(B)** RFP intensity increases when GPLD1-RFP-GFP is incubated with SAA1. Protein of interest-RG with or without ALFA-SAA1-N in serum-free medium (advanced DMEM/F12) was preincubated at 37°C with shaking for 2 h. HEK293 cells were treated with the medium for 18 h and analyzed by flow cytometry. **(C, D)** GPLD1-RFP-GFP is degraded in lysosomes. GPLD1-RG with or without ALFA-SAA1-N in serum-free medium was preincubated at 37°C with shaking for 2 h. HEK293 cells were treated with the media in the presence or absence of bafilomycin A$_1$ (BafA) for 18 h. **(C, D)** Cells were analyzed by flow cytometry (C) or immunoblotting using anti-mCherry and anti–$\beta$-tubulin antibodies (D). **(C)** The bar graph shows relative fluorescence intensities normalized to non-treated cells (n = 3) (C). **(D)** The relative band intensity is the –fold change of each sample's intensity, normalized to the intensity of ALFA-SAA1-N treated cells, which was set to 1.0 (n = 3) (D). Data are presented as mean ± SEM. Exact P-values are also shown above the bars.

## GPLD1 binds aberrant proteins

To investigate whether GPLD1 directly interacts with substrate proteins, we conducted immunoprecipitation assays using recombinant proteins. Denatured ALFA-SAA1-N or native ALFA-SAA1-F (full-length) was incubated with GPLD1-RG, followed by immunoprecipitation using ALFA nanobody-conjugated beads. GPLD1 was found to co-immunoprecipitate with ALFA-SAA1-N and -F (Fig 2A). To assess binding in the context of protein misfolding, we used luciferase as a model substrate as it undergoes heat-induced denaturation at 42°C (Glover & Lindquist, 1998; Rodrigo-Brenni et al, 2014). Immunoprecipitation revealed that the association between GPLD1-RG and luciferase was enhanced under conditions promoting misfolding (Fig 2B). These findings demonstrate that GPLD1 directly binds aberrant and misfolded proteins.

We further tested if GPLD1 facilitates the uptake of native SAA1-F. Surprisingly, GPLD1-RG internalization was not stimulated by either native or denatured ALFA-SAA1-F (Fig S3A). Sedimentation analysis revealed that, unlike denatured SAA1-N, which readily transitioned to the pellet fraction, SAA1-F (both native and denatured) showed negligible aggregation in PBS (Fig S3B). This demonstrates the low aggregability of full-length SAA1. These data suggest that binding to GPLD1 does not, per se, trigger internalization; rather, the formation of SAA1 aggregates appears to be required for GPLD1-mediated transport to the lysosome.

We evaluated whether GPLD1 suppresses SAA1-N aggregation to determine its role as an extracellular chaperone. Although GPLD1-RG exhibited a modest inhibitory effect on aggregation, this reduction did not reach statistical significance compared with the albumin control (Fig S3C). These findings imply that instead of acting as a classic holdase chaperone, GPLD1 functions as a

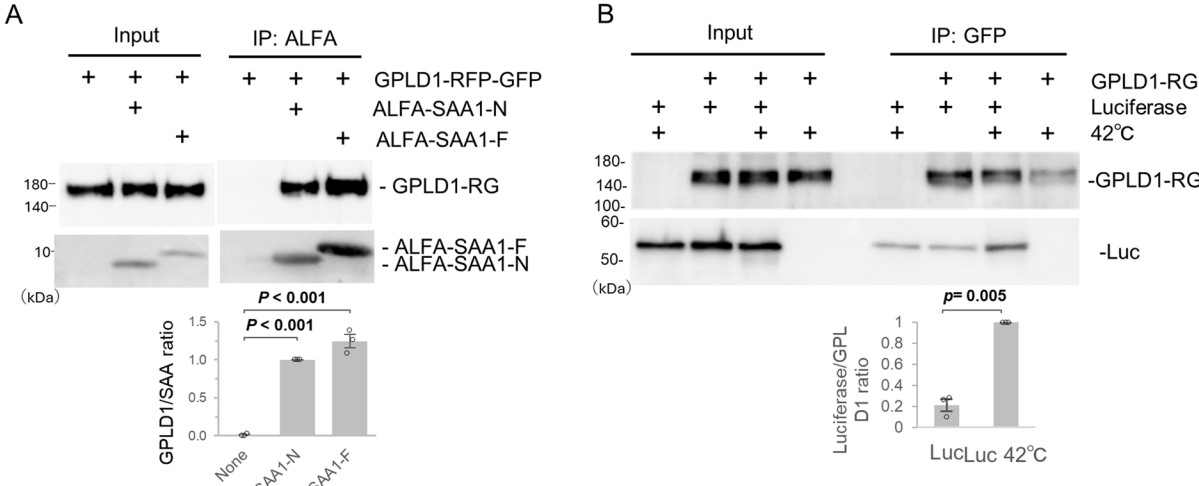

**Figure 2. GPLD1 directly interacts with aberrant proteins.**
**(A)** Recombinant GPLD1-RFP-GFP was mixed with or without ALFA-SAA1-N or -F in PBS and preincubated at 37°C with shaking for 2 h. Samples were then subjected to immunoprecipitation using ALFA-nanobody beads at 4°C. **(B)** Recombinant GPLD1-RFP-GFP was mixed with or without luciferase in PBS and incubated at 42°C for 30 min. Samples were subjected to immunoprecipitation using GFP nanobody beads at 4°C. The relative band intensity is the –fold change of each sample's intensity, normalized to the intensity of SAA1-N or Luciferase at 42°C treated cells, which was set to 1.0 (n = 3). Data are presented as mean ± SEM. Exact *P*-values are also shown above the bars.

scavenger carrier by facilitating the clearance of SAA1 aggregates via the lysosomal pathway.

### GPLD1 and the aberrant protein complex are degraded by lysosomes

To investigate lysosomal degradation mediated by GPLD1, we conducted an internalization assay using serum-free medium containing GPLD1-RG and various substrate proteins. S-formylglutathione hydrolase (ESD), a substrate for the extracellular chaperones clusterin and α2-macroglobulin (α2M), has also been identified as a misfolded protein substrate (Tomihari et al, 2023). GPLD1-RG was mixed with luciferase or ALFA-tagged ESD (ALFA-ESD) in serum-free medium. The luciferase mixture was denatured at 42°C for 30 min, whereas the ALFA-ESD mixture was denatured at 50°C for 1 h to induce misfolding. After replacing the cell culture medium with medium containing the GPLD1-RG–substrate mixtures, cells were incubated overnight and subsequently analyzed by flow cytometry. Compared with the GPLD1-RG–only controls, the addition of luciferase or ALFA-ESD significantly increased RFP intensity (Fig 3A). As negative control experiments, heat stress alone (at either 42°C or 50°C) had no effect on the internalization of GPLD1-RG (Fig S4A). Furthermore, GPLD1-RG is not coprecipitated with denatured luciferase, SAA1-N, ESD, or heat stress (Fig S4B and C). These data suggest that GPLD1-RG promotes the lysosomal degradation of these misfolded proteins.

To examine the subcellular localization of GPLD1, we performed double immunostaining using a GPLD1 construct fused to Gamillus, a GFP resistant to lysosomal degradation (GPLD1-Gamillus-His) (Shinoda et al, 2018). Cells were treated as described in Figs 1B and 3A with GPLD1-Gamillus, followed by immunostaining with an anti-LAMP1 antibody, a marker of the lysosomal membrane (Fig 3B). In cells treated with GPLD1-Gamillus alone, green fluorescence was barely detectable. Conversely, the presence of luciferase or ALFA-SAA1-N markedly increased both the number and intensity of green fluorescent puncta. These GPLD1-Gamillus puncta colocalized with LAMP1-positive red signals, indicating lysosomal localization. Treatment with BafA suppressed the accumulation of RFP, but not GFP, in cells treated with GPLD1-RG and luciferase (Fig 3C). In addition, RFP cleavage assays demonstrated that BafA inhibits the increase in free RFP in cells exposed to GPLD1-RG and SAA1-N (Fig 3D). These combined cell biological and biochemical findings suggest that the GPLD1–aberrant protein complex is internalized and trafficked to the lysosomal lumen for degradation.

### Vascular endothelial cells exhibit higher levels of internalization of the GPLD1 complex

We performed GPLD1-RG internalization assays in four cell lines of distinct tissue origins: HepG2 (liver carcinoma), U2OS (osteosarcoma), T98G (glioblastoma), and HuEhT (immortalized human vascular endothelial cells). Although no significant increase was observed with luciferase in any cell type, substrate-dependent internalization of GPLD1-RG was enhanced in most cell lines, with the exception of HepG2 cells (Fig S5). Notably, the most substantial increase was detected in the vascular endothelial cell line HuEhT. In contrast, HepG2 cells exhibited no significant increase in GPLD1-RG internalization across all tested substrates. These results suggest that although GPLD1 is secreted from the liver, it mediates the degradation of extracellular proteins with a certain degree of tissue specificity.

### Catalytic activity of GPLD1 is not required for internalization of the GPLD1 complex

GPLD1 exhibits catalytic activity that specifically hydrolyzes the inositol phosphate bond of GPI-anchored proteins on the cell membrane (Cao et al, 2023). Mutations at His133 or His158 abolish this enzymatic activity (Raikwar et al, 2005). To determine whether GPLD1's catalytic

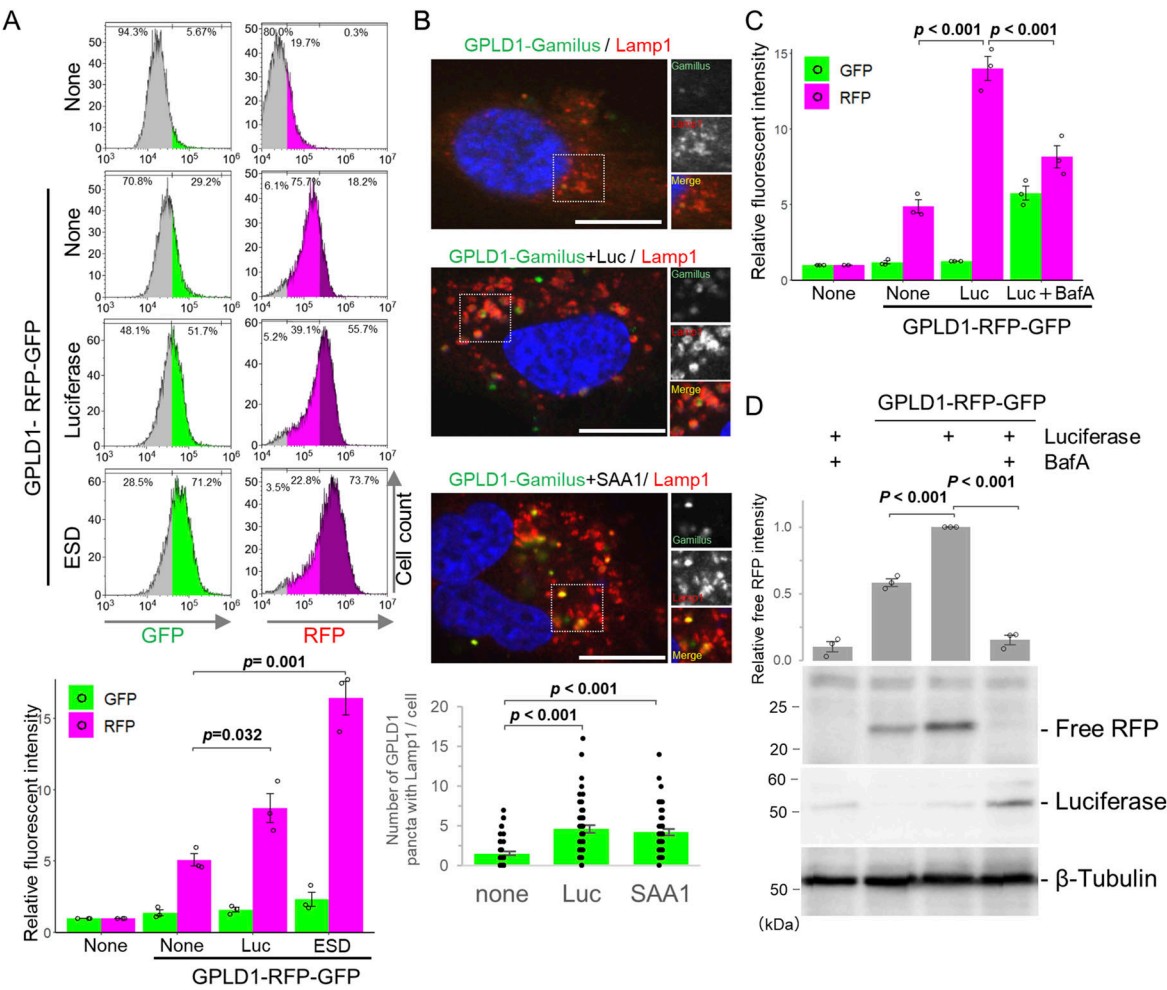

**Figure 3. GPLD1 mediates lysosomal degradation of misfolded proteins.**
**(A)** GPLD1-RFP-GFP internalization is enhanced in the presence of misfolded proteins. GPLD1-RFP-GFP was preincubated either alone (50°C for 1 h), with luciferase (Luc) (42°C for 30 min), or ALFA-ESD (50°C for 1 h) in serum-free medium. HEK293 cells were cultured with medium containing GPLD1-RFP-GFP with or without substrate for 18 h and analyzed by flow cytometry. **(B)** GPLD1 is transported into lysosomes. GPLD1-Gamillus was preincubated with luciferase (42°C for 30 min) or ALFA-SAA1-N (SAA1) (37°C with shaking for 2 h) in serum-free medium. HuEhT cells were treated with the medium for 18 h. Cells were immunostained for LAMP1 (lysosomal marker) and imaged by confocal microscopy. Scale bar, 10 $\mu$m. **(C, D)** GPLD1-RFP-GFP and substrate complexes are degraded in lysosomes. **(A, C, D)** HEK293 cells were treated as in (A) in the presence or absence of bafilomycin A$_1$, then analyzed by flow cytometry (C) or immunoblotting with anti-mCherry, anti-luciferase, and anti–$\beta$-tubulin antibodies (D). **(A, C)** The bar graph shows relative fluorescence intensities per cell normalized to untreated controls (n = 3) (A, C). **(B)** The number of GPLD1 puncta co-localizing with Lamp1 per cell is shown (n = 50) (B). **(D)** The relative band intensity is the –fold change of each sample's intensity, normalized to the intensity of luciferase-treated cells, which was set to 1.0 (n = 3) (D). Data are presented as mean ± SEM. Exact *P*-values are also shown above the bars.

function is required for substrate-dependent intracellular uptake, we generated two catalytically inactive mutants—GPLD1-H133N-RG and GPLD1-H158N-RG—by substituting histidine residues at positions 133 and 158 with asparagine (Fig S6). Internalization assays revealed that these mutants did not significantly reduce RFP accumulation compared with WT GPLD1 (Fig 4A). Given that these mutations effectively eliminate enzymatic activity (Horowitz et al, 2020), yet internalization still occurred, these findings indicate that GPLD1's catalytic activity is not essential for substrate-mediated uptake. To further investigate whether GPI-anchored proteins are involved in GPLD1 internalization, we generated GPAA1 KO cells. GPAA1 is a critical GPI transamidase required for the synthesis of GPI-anchored proteins (Eisenhaber et al, 2014). Western blot analysis confirmed the absence of GPAA1 protein in KO cells

(Fig 4B), and trafficking of GFP-tagged GPI-anchored proteins to the plasma membrane was disrupted in these cells (Fig 4C), confirming the loss of surface GPI-anchored proteins. Nevertheless, GPLD1-RG internalization was not impaired in GPAA1 KO cells, as confirmed by flow cytometry and RFP cleavage assays, respectively (Fig 4D and E), suggesting that uptake of the GPLD1–substrate complex occurs independently of GPI-anchored proteins.

### Cell surface HS facilitates GPLD1 internalization

Nearly all vertebrate cells express a limited set of cell-surface HS proteoglycans, which serve as receptors for various growth factors

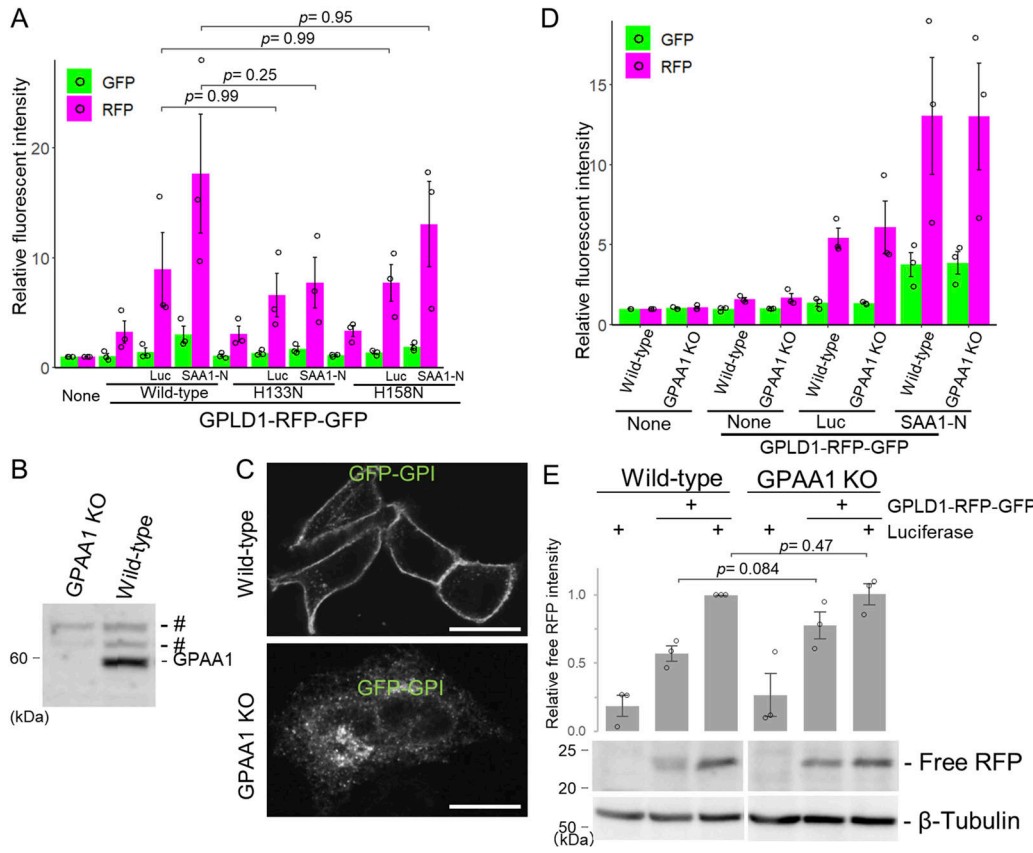

**Figure 4. GPLD1 internalization is independent of GPI-anchored protein cleavage.**
**(A)** Catalytic activity of GPLD1 is not required for its internalization. WT GPLD1-RFP-GFP or catalytic mutants (H133N, H158N) were preincubated with luciferase (42°C for 30 min) or ALFA-SAA1-N (37°C with shaking for 2 h) in serum-free medium. HEK293 cells were treated with medium containing GPLD1-RFP-GFP with or without substrate for 18 h and analyzed by flow cytometry. **(B)** Confirmation of GPAA1 KO by immunoblotting WT and GPAA1 KO HeLa cell lysates with anti-GPAA1 antibody. # indicates non-specific signals. **(C)** GPAA1 KO cells fail to mature GPI-anchored proteins. WT and GPAA1 KO HeLa cells expressing GFP-GPI were analyzed by confocal microscopy. Scale bar, 10 μm. **(D, E)** GPLD1 internalization is not impaired in GPAA1 KO cells. **(A, D, E)** WT and GPAA1 KO HeLa cells were treated as in (A) and analyzed by flow cytometry (D) or immunoblotting with anti-mCherry and anti–β-tubulin antibodies (E). **(A, D)** The bar graph shows relative fluorescence intensity per cell normalized to untreated controls (n = 3) (A, D). **(E)** The relative band intensity is the –fold change of each sample's intensity, normalized to the intensity of luciferase-treated cells, which was set to 1.0 (n = 3) (E). Data are presented as mean ± SEM. Exact P-values are also shown above the bars.

and viruses (Yayon et al, 1991; Liu & Thorp, 2002; Xu & Esko, 2014; Pillay et al, 2016). In addition, HS functions as a membrane receptor for the substrate-dependent internalization of clusterin (Itakura et al, 2020). To investigate whether HS is similarly involved in the substrate-dependent uptake of GPLD1, we first examined the potential direct interaction between GPLD1 and HS. A pulldown assay was performed using heparin-coated beads, which mimic the structure of HS chains. GPLD1-RG, but not the RG tag alone, was precipitated by the heparin-coated beads (Fig 5A), indicating a specific interaction. Furthermore, this binding was competitively inhibited by the addition of free heparin.

We conducted GPLD1-RG internalization assays using EXT1 KO cells, which lack EXT1, an essential enzyme for the biosynthesis of HS (McCormick et al, 1998). In these cells, the luciferase-induced increase in RFP levels was reduced to less than half compared with WT cells and EXT1 overexpression (EXT1 OE) cells (Fig 5B). Notably, over-expression of EXT1 in KO (EXT1 OE-EXT1 KO) cells restored RFP intensity, suggesting that HS biosynthesis is critical for GPLD1-RG internalization. To further examine the direct role of extracellular HS in this

process, we performed a competitive inhibition assay using free heparin. GPLD1-RG was mixed with luciferase or ALFA-SAA1-N in serum-free medium and incubated with cells in the presence or absence of excess heparin. The addition of heparin significantly reduced GPLD1-RG internalization, lowering uptake to less than half of the original level (Fig 5C). Collectively, these results indicate that cell-surface HS mediates substrate-dependent internalization of GPLD1.

## Substrate selectivity of GPLD1 differs from that of clusterin

To compare the substrate selectivity of GPLD1 with that of clusterin, we performed internalization assays using clusterin-RG and GPLD1-RG at equal concentrations. Three types of substrate proteins were tested: luciferase, ALFA-SAA1-N, and amyloid β 1–42 (Aβ42)-ALFA. Clusterin-RG induced the highest increase in RFP intensity in the presence of Aβ42, whereas its response to SAA1-N was comparatively lower. Conversely, GPLD1-RG showed a greater increase in RFP intensity with SAA1-N than with Aβ42 (Fig 6A). To determine if substrate-selective uptake correlates with binding,

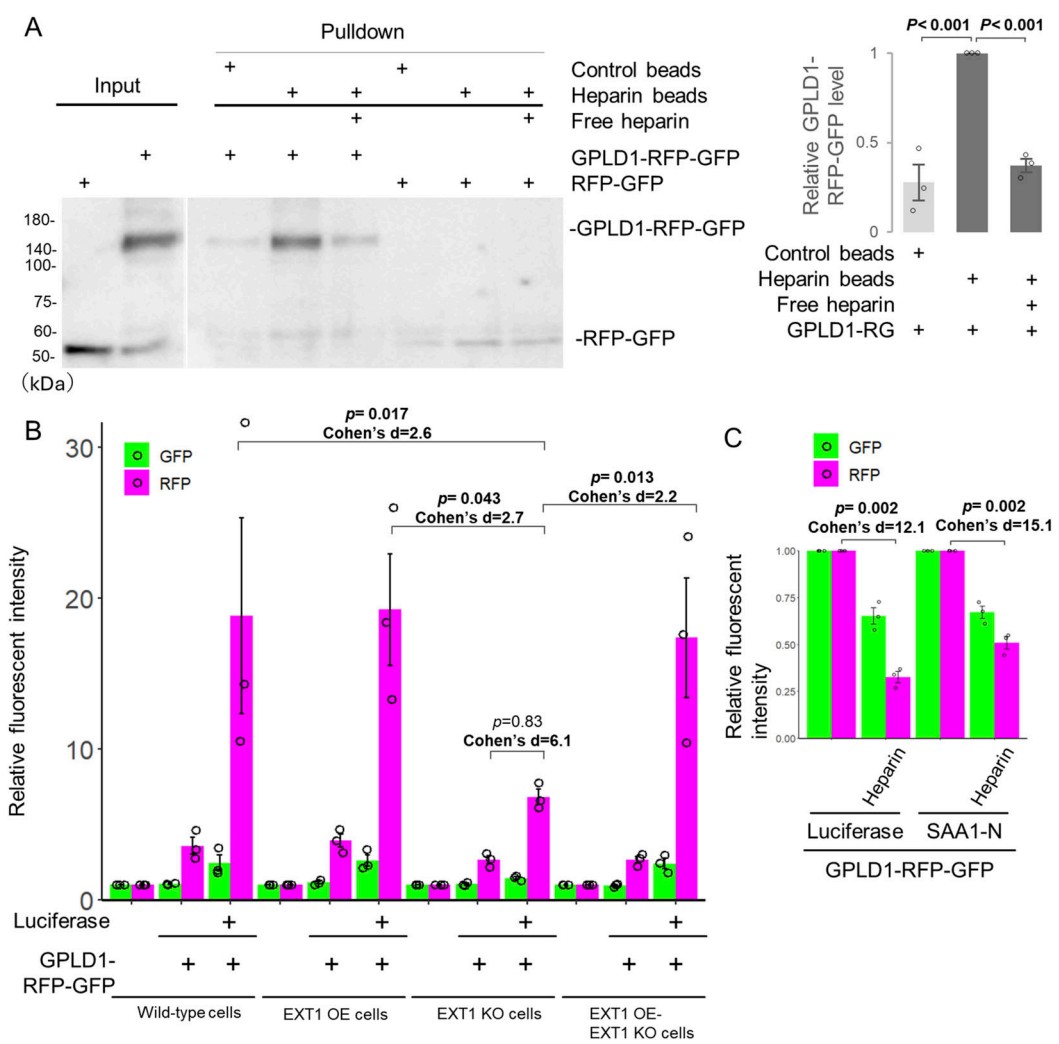

**Figure 5. Internalization of GPLD1 depends on heparan sulfate (HS).**

**(A)** GPLD1 directly binds HS. Recombinant GPLD1-RFP-GFP or RFP-GFP, in the presence or absence of free heparin, was subjected to a pulldown assay using heparin-conjugated beads. The relative band intensity of GPLD1-RG is the –fold change of each sample's intensity (n = 3). **(B)** EXT1 KO cells show reduced GPLD1-RFP-GFP internalization. GPLD1-RFP-GFP was preincubated with luciferase (42°C for 30 min) in serum-free medium. HEK293 WT, EXT1 KO cells complemented with EXT1 (overexpression, OE), or empty vector were cultured in medium containing GPLD1-RFP-GFP with or without luciferase for 18 h and analyzed by flow cytometry. The bar graph shows relative fluorescence intensity per cell normalized to untreated cells (n = 3). Data are presented as mean ± SEM. Exact *P*-values and effect sizes (Cohen's d) are also shown above the bars. **(C)** Excess free heparin inhibits GPLD1-RFP-GFP internalization. GPLD1-RFP-GFP was incubated with luciferase (42°C for 30 min) or ALFA-SAA1-N (37°C with shaking for 2 h) in serum-free medium. HEK293 cells (for ALFA-SAA1-N) or HuEhT cells (for luciferase) were treated with medium containing GPLD1-RFP-GFP in the presence or absence of heparin for 18 h and analyzed by flow cytometry. The bar graph shows relative fluorescence intensity per cell normalized to the non-heparin–treated control (n = 3). Data are presented as mean ± SEM. Exact *P*-values and effect sizes (Cohen's d) are also shown above the bars.

immunoprecipitation assays were performed. We found that GPLD1 preferentially binds to SAA1-N over Ab42, whereas clusterin showed the opposite binding profile (Fig 6B). These results suggest that GPLD1 exhibits a substrate selectivity profile distinct from that of clusterin.

## Discussion

We previously identified the CRED pathway, in which extracellular chaperones, such as clusterin bind aberrant proteins, forming complexes that are selectively internalized via specific cell-surface receptors and subsequently degraded in lysosomes, thereby facilitating the removal of aberrant proteins from the extracellular space (Itakura et al, 2020; Tomihari et al, 2023). In the present study, we identified GPLD1 as a novel extracellular scavenger carrier. Co-immunoprecipitation of purified GPLD1 demonstrated its ability to directly interact with SAA1-N and misfolded proteins (Fig 2). Internalization assays using GPLD1-RG showed that GPLD1 mediates the cellular uptake of both SAA1-N and misfolded proteins (Fig 3A). We further confirmed that this internalization leads to lysosomal degradation, as evidenced by fluorescence microscopy and RFP cleavage assays (Fig 3B and C). In addition, our data indicate that

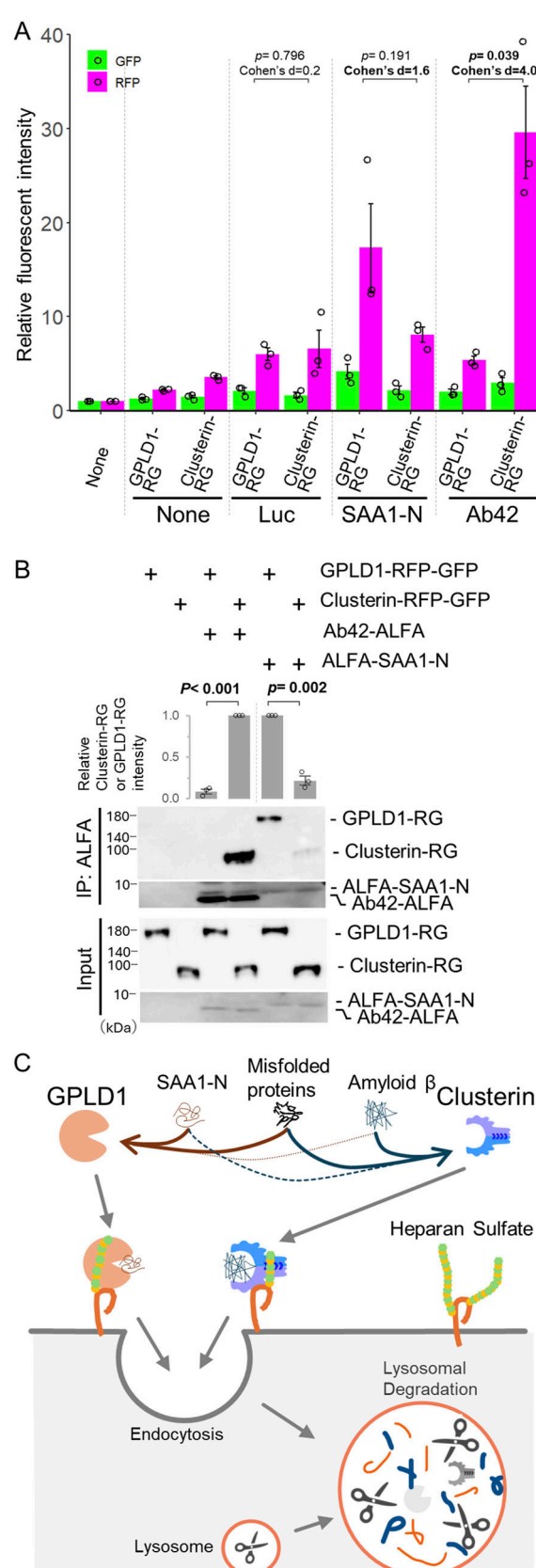

**Figure 6. GPLD1 and clusterin exhibit different specificities.**
**(A)** GPLD1, but not clusterin, shows higher specificity for SAA1. GPLD1-RFP-GFP (0.2 $\mu$M) or clusterin-RFP-GFP (0.2 $\mu$M) was incubated with 0.2 $\mu$M luciferase (42°C for 30 min), 8 $\mu$M ALFA-SAA1-N (37°C with shaking for 2 h), or 8 $\mu$M amyloid $\beta$1-42 (Ab42) (37°C with shaking for 2 h) in serum-free medium. HuEhT cells were treated with the medium containing GPLD1-RFP-GFP or clusterin-RFP-GFP, with or without substrate, for 18 h and analyzed by flow cytometry. The bar graph shows relative fluorescence intensity per cell normalized to non-treated controls (n = 3). **(B)** Differential substrate preferences of GPLD1 and clusterin. Recombinant GPLD1-RFP-GFP or clusterin-RFP-GFP was mixed with or without Luc, ALFA-SAA1-N, or ALFA-Ab42 in PBS and preincubated at 37°C with shaking for 2 h. Samples were then subjected to immunoprecipitation using ALFA-nanobody beads at 4°C. The relative band intensity is the –fold change of each sample's intensity, normalized to the intensity of clusterin-RG with Ab42-ALFA or GPLD1-RG with ALFA-SAA1-N, which was set to 1.0. Data are presented as mean ± SEM. Exact P-values are also shown above the bars (n = 3). Data are presented as mean ± SEM. Exact P-values and effect sizes (extn's d) are also shown above the bars. **(C)** Model of the extracellular aberrant protein degradation pathway by GPLD1. GPLD1 interacts with misfolded proteins and SAA1, and the GPLD1 complex is delivered to lysosomes via cell surface heparan sulfate.

this process is dependent on cell-surface HS (Fig 5). Collectively, these findings suggest that GPLD1 facilitates the selective lysosomal degradation of aberrant protein complexes through an HS-dependent mechanism at the cell surface (Fig 6C).

SAA family proteins, including SAA1, are known to be up-regulated in chronic inflammatory conditions such as metabolic syndrome, diabetes, and rheumatoid arthritis (Wilson et al, 2018). Elevated SAA1 expression has also been reported in the livers of patients with non-alcoholic fatty liver disease (Jiang et al, 2022). Similarly, GPLD1 expression is significantly increased in individuals with obesity, diabetes, and non-alcoholic fatty liver disease (Cao et al, 2023). Considering the lysosomal uptake activity of GPLD1 demonstrated in this study, it is plausible that GPLD1 is up-regulated to facilitate the degradation of elevated SAA1 during inflammatory responses. Although this study focused on amyloidosis-associated proteins to identify GPLD1, our findings show that GPLD1 is also effective in targeting misfolded proteins beyond amyloidogenic substrates. Therefore, GPLD1 may have potential therapeutic applications across a broad spectrum of protein misfolding-related diseases.

A comparison of substrate selectivity between GPLD1 and clusterin revealed that GPLD1 exhibits higher uptake activity for SAA1-N than clusterin (Fig 6A), suggesting that SAA1 is a specific substrate of GPLD1. GPLD1 is primarily secreted by the liver and is present in the bloodstream at relatively high concentrations (~40–160 $\mu$g/ml) (Qin et al, 2016; Bjelosevic et al, 2017). This high level may reflect its role in rapidly clearing aberrant proteins that arise in circulation. Although the liver is the main source of GPLD1 secretion, it is also expressed in the brain (Cao et al, 2023). The presence of GPLD1 in the brain suggests a potential role in the uptake of amyloidosis-associated proteins, which may help prevent Alzheimer's disease. Supporting this, increased GPLD1 expression has been reported to reduce A$\beta$42 levels through interactions involving Galectin-3 binding protein (Seki et al, 2020), indicating an indirect connection between GPLD1 and amyloid pathology.

We investigated whether HS, known to serve as a cell surface receptor for clusterin, also plays a role in GPLD1 internalization. In

EXT1 KO cells, which lack HS chains (McCormick et al, 1998), substrate-dependent uptake of GPLD1 was significantly reduced (Fig 5B). In addition, the presence of heparin in the culture medium, which mimics HS chains, decreased GPLD1 internalization (Fig 5C). However, internalization was not completely abolished in EXT1 KO cells (Fig 5B), indicating that although HS chains are not essential receptors, they likely facilitate GPLD1 uptake at the cell surface. It is plausible that HS chains, because of their extended polysaccharide structure, tether extracellular GPLD1 and may transfer it to an as-yet unidentified cell surface receptor. Indeed, HS chains function as receptors for FGF2, enhancing its binding to the FGF2 receptor (Xu & Esko, 2014; Nadanaka & Kitagawa, 2018). HS chains are a major constituent of the glycocalyx, which is highly abundant in vascular endothelial cells (Reitsma et al, 2007). Consistent with this, our data demonstrated that GPLD1-RG internalization was highest in the vascular endothelial cell line, HuEhT (Fig S5). Although heparan sulfate serves as a primary tethering factor, its relative abundance across different tissues may act as a key regulatory mechanism for determining the tissue-specific uptake of the GPLD1-substrate complex. Although the GPLD1-substrate complex is selectively internalized upon substrate binding, the mechanism by which endocytosis is preferentially promoted for this complex rather than GPLD1 alone remains unclear. We propose that binding of aberrant proteins induces a conformational change in GPLD1, enabling interaction with an unknown cell surface receptor that mediates internalization. Future studies aimed at identifying this receptor will be crucial to fully understand GPLD1-mediated uptake.

GPLD1 exhibits enzymatic activity that cleaves the lipid-binding site of GPI-anchored proteins exposed on the cell surface, thereby releasing them from the membrane (Cao et al, 2023). These released GPI-anchored proteins are thought to influence chronic diseases by modulating various metabolic pathways (Cao et al, 2023). To explore the relationship between GPLD1's enzymatic activity and its internalization, we generated catalytic mutants and conducted internalization assays. The GPLD1 H133N and H158N mutants, which display nearly undetectable enzymatic activity in biochemical assays (Raikwar et al, 2005), retained their ability to undergo internalization (Fig 4A), indicating that enzymatic activity is not essential for this process. In addition, internalization assays using cells deficient in GPI-anchored proteins confirmed that GPLD1 internalization occurs independently of these proteins (Fig 4B). Notably, GPLD1's enzymatic function has been shown to be required for cognitive improvements in mice (Horowitz et al, 2020). Thus, GPLD1 appears to be a bifunctional molecule, both cleaving GPI-anchored proteins and facilitating the degradation of aberrant proteins. Given that the accumulation of aberrant proteins is linked to impaired brain function (Currais et al, 2017), GPLD1 may contribute to neural health through a synergistic mechanism involving both its enzymatic activity and its role in aberrant protein internalization.

Molecular chaperones were originally defined as factors that assist in protein folding. Conversely, extracellular chaperones function primarily to maintain the solubility of abnormal proteins by binding them directly and preventing aggregation in the extracellular environment (Wyatt et al, 2013). Given the low ATP concentration outside the cell, extracellular chaperones lack refolding activity. Indeed, extracellular chaperones such as

clusterin do not possess ATPase domains (Poon et al, 2000) but effectively inhibit stress-induced protein aggregation (Humphreys et al, 1999). Whether GPLD1 can similarly prevent aggregate formation remains to be determined. Based on current evidence, GPLD1 functions as a scavenger carrier that facilitates the lysosomal degradation of aberrant proteins in the extracellular space. It exhibits distinct substrate selectivity from clusterin and operates within the CRED pathway (Fig 6B). Further investigation is needed to elucidate the molecular mechanisms underlying GPLD1-mediated degradation of extracellular aberrant proteins, which may inform novel therapeutic strategies for their removal. Recently, several lysosome-targeting chimeras (LYTACs) have been developed to artificially degrade extracellular substrates by using bifunctional molecules that bind both the plasma membrane and the target protein (Banik et al, 2020; Kim et al, 2024). However, because these chimeras are transported to lysosomes even in the absence of substrate, repeated dosing may be required over time. Conversely, administration of scavenger factors such as GPLD1 and clusterin may prove more efficient, as substrate binding triggers their lysosomal transport. Therefore, a thorough understanding of the substrate recognition mechanism of GPLD1 is critical to effectively target specific extracellular substrates. Collectively, our findings provide valuable insights into extracellular proteostasis and advance the field of selective degradation of extracellular aberrant proteins.

# Materials and Methods

### Cell culture

Flp-In T-Rex HEK293 cells (cat# R78007; Thermo Fisher Scientific), HepG2 cells (cat# RCB1648; RIKEN BRC), and U2OS cells were cultured in DMEM (Nacalai Tesque). T98G cells (cat# IFO50303; JCRB) were maintained in Eagle's MEM (Nacalai Tesque). HuEhT-2 cells (cat# JCRB1459; JCRB) were cultured in MCDB131 medium (Thermo Fisher Scientific) supplemented with 0.03 g/liter endothelial cell growth supplement, 5 µg/ml heparin, and 10 mM L-glutamine. All media were further supplemented with 10% FBS (Biosera) and 50 µg/ml penicillin–streptomycin. Cells were maintained in a humidified incubator at 37°C with 5% $CO_2$. To generate stable cell lines with doxycycline-inducible expression of secreted proteins (e.g., GPLD1-mCherry-sfGFP-His), Flp-In T-Rex HEK293 cells were co-transfected with plasmids encoding the proteins of interest and pOG44, which encodes FLP recombinase. Transfected cells were selected and maintained in a medium containing 100 µg/ml hygromycin. Gene expression at the FRT site was induced by treatment with 200 ng/ml doxycycline (Clontech).

### Plasmids

To construct GPLD1-mCherry-sfGFP-His (GPLD1-RG) and GPLD1-Gamillus-His (GPLD1-Gamillus), full-length human GPLD1 was amplified from total cDNA extracted from HepG2 cells by PCR. Gamillus was amplified from pcDNA3-Gamillus (a gift from Dr. Takeharu Nagai, Osaka University) (Shinoda et al, 2018). The

amplified cDNAs were cloned into the pcDNA5 FRT TO vector along with sequences encoding mCherry, sfGFP, Gamillus, and a His-tag to generate pcDNA5 FRT TO GPLD1-RG and GPLD1-Gamillus constructs. GPLD1 mutants (H133N, H158N) were generated via site-directed mutagenesis. The pcDNA5 FRT TO clusterin-mCherry-sfGFP-His (clusterin-RG) construct was described previously (Itakura et al, 2020). The pOG44 plasmid was used for the Flp-In recombination system. For the generation of His-ALFA-SAA1-N terminus (SAA1-N) (human SAA1 19–94 aa), His-ALFA-SAA1-Full length (SAA1-F) (human SAA1 19–122 aa), and Ab42 (amyloid β1–42)-ALFA-His, SAA1, and Ab42 sequences were synthesized as gBlock Gene Fragments (Integrated DNA Technologies). These fragments were inserted into the pRSET-A vector along with ALFA-tag and His-tag sequences to produce pRSET-A His-ALFA-SAA1-N, His-ALFA-SAA1-F, and pRSET-A Ab42-ALFA-His. To construct pCW SS-RFP-GFP-GPI, the signal sequence of prolactin, mCherry, sfGFP, and a 78-bp GPI–anchor sequence derived from prion protein were subcloned into the pCW57.1 vector (cat# 41393; Addgene). Lentiviral packaging plasmids pCMV-VSVG (cat# 8454; Addgene) and psPAX2 (cat# 12260; Addgene) were used for lentivirus production.

## Generation of KO cells

EXT1 KO cells were generated as previously described (Itakura et al, 2020). To create GPAA1 KO cells, a GPAA1-targeting sgRNA (CCTCTT ACCGGTGGGTATTG) was cloned into lentiCRISPRv2 hygro (cat# 98291; Addgene). HeLa cells were infected with lentivirus carrying Cas9 and the GPAA1 sgRNA. After 24 h, cells were selected with 100 µg/ml hygromycin for over 7 d, followed by cloning via limiting dilution.

## Antibodies

Rabbit polyclonal anti-LAMP1 antibody was kindly provided by Y. Tanaka (Kyushu University, Fukuoka, Japan). Rabbit polyclonal antibodies against sfGFP, mCherry, and the ALFA tag were generated by Eurofins using full-length mCherry protein and the ALFA peptide as antigens. Rabbit polyclonal anti-luciferase antibody (cat# PM016) was purchased from MBL. Rabbit polyclonal anti-β-actin (cat# 20536-1-AP) and anti-GPAA1 (cat# 10104-1-AP) antibodies were obtained from Proteintech. HRP-conjugated anti-rabbit antibody (cat# 7074S) was purchased from Cell Signaling Technology.

ALFA nanobody beads were prepared by conjugating ALFA nanobody protein (Götzke et al, 2019), purified from pRSET-A ALFA-His, to N-hydroxysuccinimide (NHS)-activated Sepharose 4 Fast Flow (GE). Similarly, GFP nanobody beads were prepared by conjugating GFP nanobody protein, purified from the pOPINE GFP nanobody plasmid (#49712; Addgene), to NHS-activated Sepharose 4 Fast Flow (GE).

## Recombinant proteins from *E. coli*

Recombinant luciferase was purchased from Promega (cat# E1701). His-ALFA-SAA1 and Ab42-ALFA-His were cloned into the pRSET-A vector and expressed in *E. coli* BL21(DE3) pLysS (cat# L1191; Promega). After harvesting, *E. coli* cells were disrupted by sonication. After ultracentrifugation at 200,000$g$ for 30 min at 4°C, the inclusion bodies (pellet fraction) containing His-ALFA-SAA1-N or Ab42-ALFA-His were solubilized by sonication in urea buffer (8 M urea, PBS, 10 mM imidazole). To remove insoluble debris, the soluble fraction was subjected to ultracentrifugation at 200,000$g$ for 30 min at 4°C. The resulting supernatant was purified using Co-NTA affinity chromatography under denaturing conditions (8 M urea, PBS). After elution using 200 mM imidazole buffer, purified His-ALFA-SAA1-N and Ab42-ALFA-His were stored in the presence of urea. His-ALFA-ESD was described previously (Tomihari et al, 2023).

## Recombinant proteins from mammalian cells

For secreted proteins from mammalian cells, pcDNA5 FRT TO plasmids encoding GPLD1-RG-His, GPLD1-Gamillus-His, and clusterin-RG-His were integrated into the FLP site of the genome in Flp-In T-Rex293 cells to generate stable (Tet-on) cell lines. Purification of secreted proteins from conditioned medium was performed as previously described (Tomihari et al, 2021). Briefly, cells were cultured for 4 d in serum-free Advanced DMEM/F12 medium (cat# 12634010; Thermo Fisher Scientific) supplemented with L-glutamine and 0.2 µg/ml doxycycline. The culture supernatant was collected and centrifuged at 1,100$g$ for 10 min at 4°C to remove dead cells. Secreted proteins were purified from the conditioned medium using Ni-NTA affinity chromatography. After elution using 200 mM imidazole buffer, the purified proteins were stored in PBS containing 10% glycerol.

## Internalization assay

Purified GPLD1-RG WT or mutant (0.2 µM) or clusterin-RG (0.2 µM) in serum-free Advanced DMEM/F12 medium or serum-free conditioned medium containing secreted GPLD1-RG was mixed with substrate proteins. It should be noted that although Advanced DMEM/F12 is serum-free, it is supplemented with specific proteinaceous growth factors, including albumin (400 mg/liter), insulin (10 mg/liter), and transferrin (7.5 mg/liter). The mixtures were incubated as follows: luciferase (0.2 µM) was heat-treated at 42°C for 30 min; His-ALFA-ESD (8 µM) was heat-treated at 50°C for 1 h; and His-ALFA-SAA1 (8 µM) and Ab42-ALFA-His (8 µM) were incubated at 37°C with shaking for 2 h. After incubation, protein aggregates were removed by centrifugation at 20,000$g$ for 5 min. Cells were then cultured with the resulting supernatant (serum-free medium containing extracellular protein-RG with or without substrate) at 37°C for 18 h. In specified experiments, 0.1 µM BafA (LC Laboratories) or 80 µg/ml heparin (Nacalai Tesque) was added together with the medium. Cells were detached using trypsin and subsequently analyzed by flow cytometry and immunoblotting. For immunostaining, cells were pre-cultured on glass coverslips.

## Flow cytometry

Cells were detached using trypsin and resuspended in a solution containing 5% newborn calf serum and 1 µg/ml 4′,6-diamidino-2-phenylindole (DAPI) in PBS. The cell suspension was then passed through a 70 µm cell strainer and collected into a 1.5 ml tube. The

collected cells were analyzed by flow cytometry using a CytoFLEX S flow cytometer (Beckman Coulter) equipped with 375-nm (DAPI), 488-nm (sfGFP), and 561-nm (mCherry) lasers. Cells were gated sequentially: first by FSC-A/SSC-A for the target population, then by SSC-W/SSC-A to select singlets. DAPI staining was used to exclude dead cells. Green and red fluorescence intensities were then measured in the remaining live singlet cells, capturing a total of 10,000 events per sample. The mean fluorescence intensity was used for analysis. Raw fluorescence intensity values obtained from flow cytometry are provided in Table S2.

### Immunostaining and fluorescence microscopy

HuEhT-2 cells cultured on coverslips were fixed at room temperature for 15 min with 3.7% formaldehyde in PBS. For immunostaining, the fixed cells were permeabilized for 5 min with 50 $\mu$g/ml digitonin in PBS. After blocking with 10% newborn calf serum in PBS for 30 min, the cells were incubated with the primary antibody for 1 h. After three washes with PBS, cells were incubated for 1 h with Alexa Fluor 568-conjugated goat anti-rabbit IgG antibody (Thermo Fisher Scientific). The cells were then mounted with a DAPI/DABCO/Glycerol/PBS solution. Stained cells were observed using a confocal laser scanning microscope (FV1000IX81; Olympus) equipped with a 100× oil immersion lens (NA 1.40).

### Cell lysis and immunoblotting

Cells were washed with cold PBS and lysed for 15 min at 4°C in lysis buffer containing 1% Triton X-100, 50 mM Tris–HCl (pH 7.5), 1 mM EDTA, and 150 mM NaCl, supplemented with an EDTA-free protease inhibitor cocktail (Nacalai Tesque) and 1 mM PMSF. After centrifugation at 20,000$g$ for 5 min, the supernatant was collected, mixed with 6× SDS sample buffer, and boiled at 95°C for 5 min. Proteins were separated by SDS–PAGE, transferred to a polyvinylidene difluoride membrane (cat# IPVH00010; Millipore), and probed with antibodies diluted in Signal Enhancer HIKARI (cat# 02270-81; Nacalai Tesque). Protein bands were visualized using ImmunoStar Zeta (cat# 297-72404; Fujifilm Wako Pure Chemical Industries).

### Identification of SAA1-binding proteins using mass spectrometry

ALFA-SAA1 (8 $\mu$M) was added to 50% bovine plasma (cat# D500-06-0500; Rockland Immunochemicals) (in PBS) and incubated for 16 h at 4°C. The plasma was then centrifuged at 20,000$g$ for 10 min to remove debris. The supernatant was mixed with ALFA-nanobody beads and rotated for 1 h at 4°C. The beads were washed four times with PBS, transferred to a new tube, mixed with SDS sample buffer, and boiled for 5 min. Bound proteins were subjected to SP3-based peptide preparation and identified by LC–MS/MS proteomics using an LTQ-Orbitrap Velos Pro (Thermo Fisher Scientific) with data-dependent MS/MS acquisition (Toh et al, 2022). Raw data were searched against the *Bos taurus* UniProt database (UP000009136, 2022_01) and cRAP using Proteome Discoverer 2.5 with the MASCOT v2.6 search engine at 1% FDR. Up to two missed cleavages were allowed. Carbamidomethylation of cysteine was set as a fixed modification, and methionine oxidation as a variable modification.

### Immunoprecipitation

Purified GPLD1-RG (final concentration: 10 nM for GFP IP, 40 nM for ALFA IP) and substrate proteins (final concentration: 40 nM luciferase for GFP IP, 500 nM His-ALFA-SAA1 for ALFA IP) were mixed in PBS and incubated at 4°C, except luciferase, which was incubated at 42°C for 30 min and His-ALFA-SAA1 at 37°C with shaking for 2 h. After incubation, excess protein aggregates were removed by centrifugation at 20,000$g$ for 5 min. The supernatant was collected; a portion was reserved as the "input" sample, whereas the remainder was mixed with GFP nanobody or ALFA nanobody beads (5 $\mu$l bed volume) and incubated at 4°C for 2 h. The beads were washed four times with PBS, transferred to a new tube, mixed with SDS sample buffer, and boiled for 5 min.

For the pulldown assay using heparin beads, 30 nM GPLD1-RG or 30 nM RG was mixed with heparin beads (cat# Super-HEP10; M&S TechnoSystems) in 0.04% BSA/PBS, in the presence or absence of 80 $\mu$g/ml free heparin, and then rotated for 2 h at 4°C. The beads were washed four times with PBS, transferred to a new tube, mixed with SDS sample buffer, and boiled for 5 min.

### Sedimentation assay

For the sedimentation assays, SAA1 (8 $\mu$M for Figs S1A and S3B), Albumin-ALFA (2 $\mu$M for Figs S1A and S3C), or GPLD1-RG (2 $\mu$M for Fig S3C) in PBS or adDMEM/F12 was incubated for 2 h at 37°C with shaking. After incubation, the mixtures were centrifuged at 20,000$g$ for 10 min at 4°C. The supernatant was collected as the supernatant fraction, whereas the remaining pellet was washed with PBS and subjected to a second centrifugation step at 20,000$g$ for 10 min. The final pellet was then collected as the pellet fraction for subsequent analysis.

### Statistical analysis

Data were collected from at least three biological replicates (n ≥ 3). One-way ANOVA with Dunnett's post hoc test was used for significance testing in Figs 1D, 2A, 3A, B, and D, 5A and B, S1C, S2C, S3A and C, S4A, and S5. One-way ANOVA with Tukey's post hoc test was used for Figs 1C, 3C, and 4A. Welch's $t$ test was applied for Figs 2B, 4E, 5C, 6A and B, and S2A and B. In addition to significance testing, effect sizes were calculated to quantify the magnitude of the observed effects. We used Cohen's d, and interpreted values based on conventional guidelines: 0.2 for a small effect, 0.5 for a medium effect, and 0.8 for a large effect for Figs 5 and 6.

## Data Availability

The LC–MS/MS data generated during this study are available in Table S1. Further information and requests for resources should be directed to and will be fulfilled by the corresponding author.

## Supplementary Information

## Acknowledgements

This work was supported by JSPS KAKENHI (grant nos. 20H03249, 22H04634, 23H04932, 24K02018 to E Itakura), the JST FOREST Program (grant no. JPMJFR204N to E Itakura), the JSPS Program for Forming Japan's Peak Research Universities (J-PEAKS) (grant no. JPJS00420230002 to E Itakura), and the Takeda Science Foundation (to E Itakura). We thank Dr. Yoshitaka Tanaka (Kyushu University) for providing the anti-LAMP1 antibodies, Dr. Takeharu Nagai (Osaka University) for the pcDNA3 Gamillus plasmid, and Dr. Reiko Nakagawa (Laboratory for Cell-Free Protein Synthesis, RIKEN Center for Biosystems Dynamics Research) for conducting the mass spectrometry experiments. We are also grateful to Akira Matsuura for his assistance with the preparation, original drafting, and critical review of the work.

### Author Contributions

M Tsuchiya: formal analysis, validation, investigation, visualization, and writing—original draft.
Y Yagishita: formal analysis and investigation.
E Itakura: conceptualization, formal analysis, supervision, funding acquisition, investigation, visualization, project administration, and writing—original draft, review, and editing.

### Conflict of Interest Statement

The authors declare that they have no conflict of interest.

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
