## [Reviewer comments · Life Science Alliance]

GPLD1 is as a scavenger carrier mediating lysosomal degradation of extracellular aberrant proteins

Mizuki Tsuchiya, Yoichiro Yagishita, and Eisuke Itakura

DOI: <https://doi.org/10.26508/lsa.202603717>

Corresponding author(s): Eisuke Itakura, Chiba University

Review Timeline:

Submission Date:	2026-03-27
Editorial Decision:	2026-05-04
Revision Received:	2026-05-09
Accepted:	2026-05-11

Scientific Editor: Tim Fessenden

Transaction Report:

Please note that the manuscript was reviewed at *Review Commons* and these reports were taken into account in the decision-making process at *Life Science Alliance*.

Reviews

Review #1

1. Evidence, reproducibility and clarity:

In this manuscript, the authors identify the protein GPLD1 as an extracellular chaperone that binds to secreted SAA1 and promotes its degradation through endocytic mechanisms. They demonstrate that GPLD1 similarly promotes degradation of the alternative substrate ESD through an activity independent of its role in cleaving GPI-anchored proteins. Further, they show that this activity depends on the receptor activity of heparan sulfate on the cell surface. These results identify GPLD1 as a new extracellular chaperone that can remove non-native proteins from extracellular environments such as the blood.

- Overall, the experiments are generally well performed, and the results are convincing. However, there are additional experiments that would better support the major finding of this manuscript identifying GPLD1 as a new extracellular chaperone that I describe below.

- Notably, additional work needs to be performed to convince that GPLD1 selectively binds non-native proteins. Is SAA1 non-native when it is bound/degraded by GPLD1? Does GPLD1 impact accumulation of amyloid A (the amyloidogenic product of SAA1)?

- Similarly, for interaction/denaturation experiments using non-native proteins, the authors generally pre-mix GPLD1 and substrate complexes prior to denaturation, which could also impact GPLD1. These experiments need to be performed where substrates are denatured prior to GPLD1 to better demonstrate the selective binding/clearance of non-native protein conformations.

- Further, the authors claim that GPLD1 has a distinct substrate profile to clusterin by primarily by monitoring clearance of SAA1 and AB1-42. It would be good to further probe this selectivity by monitoring interactions between GPLD1/clusterin and substrates in extracellular environments (e.g., blood)

- It would be good to characterize the levels of GPLD1 in in vivo extracellular environments such as the blood.

- Does addition of exogenous GPLD1 mitigate proteotoxicity of extracellular proteins in cell culture models? This is a common assay used in the identification of new extracellular chaperones and should be performed here.

- The authors should provide quantification (and replicates/stats) for all immunoblotting experiments (e.g., see Fig 2).

- Again, this is an interesting manuscript that is well written. Addressing these above comments will improve the overall findings, which will be well received by the research community.

- Minor Comments:

Please add a label for the second row of flow data in Fig. 3A (I believe it is none)

2. Significance:

The experiments are well performed and convincing. However, as indicated below, the authors should repeat immunoblots and provide replicates/stats to support their findings.

3. How much time do you estimate the authors will need to complete the suggested revisions:

Between 3 and 6 months

4. Review Commons values the work of reviewers and encourages them to get credit for their work. Select 'Yes' below to register your reviewing activity at Web of Science Reviewer Recognition Service (formerly Publons); note that the content of your review will not be visible on Web of Science.

No

Review #2

1. Evidence, reproducibility and clarity:

Summary:

In this article, the authors find that Glycosylphosphatidylinositol-specific phospholipase D1 (GPLD1) bind serum amyloid A1 and promotes its lysosomal degradation in cultured cells. This mechanism seems to be independent of cleavage GPLD1 activity and mainly dependent on heparan sulfate as a receptor. Substrate specificity appear distinct from one of the best characterized extracellular chaperones, clusterin.

Major comments:

- The authors claim that GPLD1 function as a scavenger chaperone, however no direct effect on preventing aggregate formation is addressed in the paper. The authors even mention it in the discussion, line 300-302 "Indeed, extracellular chaperones such as clusterin do not possess ATPase domains (Poon et al, 2000) but effectively inhibit stress-induced protein aggregation (Humphreys et al, 1999). Whether GPLD1 can similarly prevent aggregate formation remains to be determined". If the authors want to claim in the manuscript that GPLD1 is a chaperone, they should demonstrate it by aggregate prevention assays, for example prevention of aggregation of the serum amyloid A1 protein using Thioflavin T.

- In line with this, the authors claim that GPLD1 binds aberrant proteins; however, they do not provide evidence - particularly in the case of SAA1, for which no supporting literature is cited - that SAA1 indeed aggregates under the reported incubation conditions (2 h at 37 {degree sign}C). Moreover, in the Methods section it is stated that following incubation of GPLD1 with the substrates, the samples were centrifuged to remove aggregates, and the supernatant was subsequently added to the cells. How much of each protein remained in the supernatant versus the pellet after centrifugation? How do the authors ensure that comparable amounts of GPLD1 are being added to the cells across the different samples? A similar concern arises for the immunoprecipitation assays, is the input shown in the Western blots taken before or after centrifugation?

- Regarding the analysis of the lysosomal degradation assay, the ratio between RFP fluorescence intensity and GFP fluorescence intensity would be more informative and easier to interpret. That ratio can then be normalized by the control sample. I suggest to include that analysis for clarity at least as supplementary/additional data along with the raw fluorescence intensity values. The authors should also specify if they use the mean or the median fluorescence intensity.

- Based on the results of Fig. 2A, the authors do not use the full SAA1 protein but just some short amyloidogenic sequence. The sequence of the peptide is not specified. This should be detailed in the manuscript.

- In the western blots where the degradation of GPLD1-RG is verified biochemically (Fig. 1E and 3D) just the free RFP is shown. The authors should also show the full length GPLD1-RG (110 kDa+GFP+RFP).

- The authors state that SAA1 was purified in the presence of urea. How much is urea diluted when SAA1 is mixed with GPLD1, and what is the final concentration in the assay? Given that urea may affect the stability of GPLD1, this information is critical.

- When ALFA-ESD is used as a substrate, the GPLD1-RG/ALFA-ESD mixture is incubated at 50 {degree sign}C for 1 h. Such conditions are quite harsh and potentially denaturing for most proteins. Did the authors test whether GPLD1-RG remains soluble and in a native state under these conditions? This point is particularly important since in Fig. 3A only one control condition with GPLD1-RG (without substrate) is shown, whereas separate controls for Luc and ESD should be included, given that the incubation conditions differ (Luc: 42 {degree sign}C, 30 min; ESD: 50 {degree sign}C, 1 h). Under which condition is the single control used as reference for the statistical analysis?

- Line 173-174 "Although all cell lines showed increased internalization upon exposure to substrate proteins, the extent varied among them. HepG2 cells exhibited the lowest levels of ..." Without quantification, it is difficult to draw

firm conclusions; at first glance, HepG2 cells do not appear to show any clear increase with any of the substrates, and in the case of ESD, the signal even seems to decrease.

- The Methods section does not provide details on how mass spectrometry was performed.

- Providing more robust evidence for substrate degradation in the presence of GPLD1-RG would strengthen the claims of the paper. Optional experiments could include repeating the assay shown in Fig. 3D (Western blot of samples with/without bafilomycin) using SAA1 and ESD as substrates, as well as performing microscopy similar to Fig. 3B but labeling the substrates and analyzing multiple time points after their addition to the cells.

****Minor comments:****

- Fig. 1A. Ruby staining should be clarified as protein staining in the text or the figure since it is not as broadly used as Coomassie. Also, this panel does not seem to be informative so it could be excluded from the manuscript.

- Line 104-105 "After excluding well-characterized proteins, we shortlisted seven candidate proteins as potential extracellular chaperones". It would help the reader if the proteins are listed in the text.

- In line 67-69, AA amyloidosis and AA amyloid is mentioned, but that acronym was not previously introduced. It was mentioned just SAA.

- In line 120 "(Fig. 1C)" should be (Fig. 1 C, D).

- Incubation time, temperature and condition (shaking?) should be clearly indicated through the manuscript and in figure legends.

- Line 169-177 and Fig. S1. The fact that the internalization (red signal) and not lysosomal degradation (green and red signals) is analyzed in this experiment when GLPD1-RFP-GFP is used is confusing and not consistent with the rest of the manuscript. Why is the GFP signal not shown in this case as well as in Fig. 5C?

- Fig. 4A, the label on the y-axis is missing.

- In the relative fluorescence graphs would be useful if more and bigger numbers are shown in the y-axis.

- Fig.6A. The amount of GPLD1-RG and Clusterin-RG is indicated in the legend but not the amount of substrate. This should be indicated.

- Including quantification of the IP results in Fig. 2B, the microscopy in Fig. 3B and the lysosomal internalization/lysosomal degradation in Fig. 4D would strengthen the robustness of the results.

****Referee Cross-commenting****

All reviewers agree on the need of additional experiments, clarifications and controls to reach publication level. There are several main points that are raised by at least 2 of the reviewers like quantification of microscopy and western blot results, more clear representation of the flow cytometry data, concern about the possible denaturation of GLPD1 when incubated with the substrate (instead of incubation after substrate denaturation), the actual denaturation and specificity for misfolded substrates and the chaperone claim without showing any aggregation prevention experiments. Therefore, a major revision is needed with an estimate time of 3-6 months.

2. Significance:

General assessment: Although extracellular protein aggregation underlies several human pathologies, such as neurodegeneration and systemic amyloidosis, extracellular proteostasis factors involved in the clearance of these aberrant aggregates remain poorly characterized. Therefore, the results presented in this article identifying GPLD1 as a potential new player in this context are significant, novel, extend the knowledge in the field of extracellular proteostasis and may have clinical relevance. The data is presented in a comprehensive manner; however, further clarifications and additional analyses are needed to strengthen the robustness of the results (see 1. Evidence, reproducibility and clarity).

Advance: The manuscript describe a conceptual advance in understanding the clearance mechanisms of extracellular misfolded proteins.

Audience: The target audience is specialized in proteostasis.

My field of expertise is chaperones, protein aggregation and extracellular proteostasis.

3. How much time do you estimate the authors will need to complete the suggested revisions:

Between 1 and 3 months

4. Review Commons values the work of reviewers and encourages them to get credit for their work. Select 'Yes' below to register your reviewing activity at Web of Science Reviewer Recognition Service (formerly Publons); note that the content of your review will not be visible on Web of Science.

No

Review #3

1. Evidence, reproducibility and clarity:

****Summary:****

- This manuscript describes the identification of glycosylphosphatidylinositol-specific phospholipase D1 (GPLD1) as a novel extracellular scavenger chaperone. The authors report that, in pulldown assays from bovine plasma, GPLD1 binds the amyloidosis-associated protein Serum Amyloid A1 (SAA1). They go on to develop a fluorescence-based internalisation assay showing that GPLD1 can bind misfolded proteins, promote their uptake into cells via heparan sulphate on the cell surface, and direct them to lysosomal degradation. This function is independent of GPLD1's catalytic activity towards GPI-anchored proteins and appears to differ from the better-studied chaperone clusterin, with GPLD1 showing a particular preference for SAA1. The authors place this mechanism within the CRED pathway of extracellular protein quality control.

- Although the topic is timely and potentially important, there are a number of technical inconsistencies, missing controls and unexplained results that make the conclusions less secure. In its current form, the work leaves several key points unresolved for publication, and will require substantial additional experimentation and clarification. On that basis, I recommend major revision at this stage.

****Major comments:****

- Fig. 1C. Could the authors include the gating strategy for their FC data? It is also unclear from the legend or the M&M how the analysis was done. Why did the authors chose this visualization and not dot blots or stacked histograms? F.e. what is the difference between the bright red part of the histogram and the Bordeaux part? I would recommend using a more clear and commonly used representation.

- Fig. 1D. One would normally expect BafA treatment to increase or at least maintain the red signal, since the RFP-tagged protein should accumulate in lysosomes without being broken down (as suggested by the authors in Fig. 1E). However, in Fig. 1D, the authors observe a decrease in the RFP intensity when BafA is added, which is counterintuitive if RFP were purely tracking cargo delivery. Could the authors validate the observation using immunofluorescence microscopy with e.g. markers of endosomes?

- The latter would also be necessary to distinguish between surface-bound and internalized material. Surface quenching controls in the FC data would make the interpretation more robust.

- Fig. 1E. The bar graph is missing. We only see a blot.

- Fig. 2B. The authors did not include a non-substrate control neither a protein that does not get bound to GPLD1,

which would help confirm substrate selectivity. It is further unclear what the n-value is for this experiment.

- Fig. 3B. The images show more GPLD1-Gamillus puncta with luciferase than with SAA1, which is surprising given the emphasis on SAA1 as the physiological substrate. The authors should: (I) quantify puncta per cell and co-localization with LAMP1 across independent replicates, (II) ensure equal input amounts of substrate and include a dose-response, and (III) perform a short time-course to see whether the difference reflects kinetics, aggregate size, or selection of images. The appearance of the green dots also does not appear to be correlating with the bar graph in Fig. 3C. Of note, the legend mentions the intensities have been taken from 3 cells. It is better to perform 3 independent experiments in which each 10-15 cells are analyzed.

- Fig. 4A. The authors conclude that catalytic activity is not required for internalization, but they do not show that these mutant proteins are folded and secreted at levels comparable to wild-type (or might be stuck in the ER?). A simple western blot of the conditioned medium could confirm this.

- Fig. 4B. The KO validation blot contains 2 "#", what type of bands are these?

- Fig. 5. The figure legend and panels need clearer indication of biological replicates, exact p-values and effect sizes. For EXT1 KO the residual activity is big enough that statistical/biological replication is essential. Fig. 5C legend ... what do the authors mean with a "cell"?

- Fig. 5B. There is an incomplete abolishment of internalisation in EXT1 KO cells. This would indicate the involvement of another receptor. Can the authors discuss a bit more.

- General remark. The authors suggest GPLD1 may be an extracellular chaperone, but they only measure uptake/degradation, not whether GPLD1 keeps proteins soluble and prevents aggregation. Assessing whether GPLD1 can actually prevent aggregate formation, rather than only clear existing aggregates, would clarify its functional overlap with other chaperones.

****Minor comments:****

- While not color-blind myself, the authors could take into consideration that other readers might be and change the colors of their FC data to rather magenta-green or other combinations of colors.

2. Significance:

This work tackles an important question: how extracellular misfolded proteins are identified and removed. If convincingly demonstrated, the finding that GPLD1 participates in this process, and does so with a distinct substrate preference, would be a valuable addition to the field of extracellular proteostasis. At present, however, the conceptual advance is undermined by gaps in methodology, inconsistencies between datasets and the absence of orthogonal or in vivo validation. Moreover the potential relevance to neurodegenerative diseases might be overstated: there is a clear link of for instance clusterin (genetically and biologically) but such causality does not exist for GPLD1. The role of GPLD1 as a selective extracellular chaperone is an interesting idea, but the evidence remains incomplete and in several places not clear. In my view the current findings require further scrutiny. Considerable further work is required to reach this level.

3. How much time do you estimate the authors will need to complete the suggested revisions:

Between 3 and 6 months

4. Review Commons values the work of reviewers and encourages them to get credit for their work. Select 'Yes' below to register your reviewing activity at Web of Science Reviewer Recognition Service (formerly Publons); note that the content of your review will not be visible on Web of Science.

Yes

Response to the Reviewers: We wish to express our sincere gratitude to all three reviewers for their insightful and constructive comments. In response to your suggestions, we have performed 29 additional experiments and have either revised or added 22 panels across the figures to the manuscript. In this revised version, we have added Yoichiro Yagishita as a co-author, as he performed the additional experiments required for the revision. These extensive updates have significantly strengthened the evidence supporting our findings on the extracellular proteostasis of GPLD1 and improved the clarity of the protocol. We have addressed each reviewer's comments in a point-by-point manner below.

Reviewer #1 (Evidence, reproducibility and clarity (Required)):

In this manuscript, the authors identify the protein GPLD1 as an extracellular chaperone that binds to secreted SAA1 and promotes its degradation through endocytic mechanisms. They demonstrate that GPLD1 similarly promotes degradation of the alternative substrate ESD through an activity independent of its role in cleaving GPI-anchored proteins. Further, they show that this activity depends on the receptor activity of heparan sulfate on the cell surface. These results identify GPLD1 as a new extracellular chaperone that can remove non-native proteins from extracellular environments such as the blood.

Overall, the experiments are generally well performed, and the results are convincing. However, there are additional experiments that would better support the major finding of this manuscript identifying GPLD1 as a new extracellular chaperone that I describe below.

Notably, additional work needs to be performed to convince that GPLD1 selectively binds non-native proteins. Is SAA1 non-native when it is bound/degraded by GPLD1? Does GPLD1 impact accumulation of amyloid A (the amyloidogenic product of SAA1)?

Response: We appreciate the reviewer's insightful suggestion. While our study demonstrates that GPLD1 mediates the degradation of SAA1 aggregates, we agree that it is crucial to clarify whether GPLD1 also facilitates the degradation of native proteins.

In our previous experiments, we utilized a truncated form of SAA1 containing the N-terminal 76 amino acids (SAA1-N), a sequence highly prone to aggregation, which was denatured in the presence of urea. To address the reviewer's concern, we have now prepared full-length SAA1 (SAA1-F) in its native state and compared it with urea-denatured SAA1-F.

Our results showed that while SAA1-F also binds to GPLD1 (Fig. 2A), interestingly, neither native SAA1-F nor urea-denatured SAA1-F promoted the internalization of GPLD1-RG (Fig. S3A).

Importantly, while urea-denatured SAA1-N readily forms aggregates in PBS and is recovered in the pellet fraction after centrifugation (sedimentation assay)(Fig. S1A), SAA1-F was not detected in the pellet fraction even under denaturing conditions (Fig. S3B). These data suggest that binding to GPLD1 does not, per se, trigger internalization; rather, the formation of SAA1 aggregates appears to be required for GPLD1-mediated transport to the lysosome.

To investigate whether GPLD1 suppresses the accumulation of SAA1-N, we performed Thioflavin T (ThT) assays and sedimentation assays. Although we monitored amyloid formation via ThT fluorescence, no significant increase in signal was observed even in the presence of SAA1-N, indicating that SAA1-N did not form typical amyloid fibrils under our experimental conditions (Response Figure 1; see below). Furthermore, the sedimentation assay revealed that the presence of GPLD1-RG did not significantly reduce the amount of SAA1-N in the pellet fraction (Fig. S3C). These results suggest that GPLD1 does not possess conventional chaperone activity aimed at inhibiting protein aggregation. calling GPLD1 a "chaperone" may lead to a misunderstanding, as its ability to directly inhibit aggregate formation remains unclear from our current data. To more accurately reflect its functional role, we have decided to use the term "scavenger carrier" throughout the revised manuscript. Accordingly, we propose that GPLD1 functions primarily as a "scavenger carrier" that recognizes and targets aberrant proteins for degradation. We believe this terminology better describes its specific action in capturing non-native species and facilitating their lysosomal clearance without necessarily implying folding-related chaperone activities.

Response Figure 1. Thioflavin T assay of SAA1-N
 Urea-denatured ALFA-SAA1-N was mixed with or without 40 μ M GPLD1-Flag in PBS containing 10 μ M Thioflavin T in a 384-well plate and incubated at 37 $^{\circ}$ C with continuous shaking (300 rpm). Thioflavin T fluorescence was measured using a microplate reader at the indicated time points.

We performed the ThT assay to evaluate the aggregation of SAA1-N; however, no increase in fluorescence was observed for SAA1-N alone. The addition of GPLD1 led to a marked increase in fluorescence, suggesting that the ThT dye may be non-specifically binding to GPLD1 itself rather than indicating substrate aggregation.

Similarly, for interaction/denaturation experiments using non-native proteins, the authors generally pre-mix GPLD1 and substrate complexes prior to denaturation, which could also impact GPLD1. These experiments need to be performed where substrates are denatured prior to GPLD1 to better demonstrate the selective binding/clearance of non-native protein conformations.

Response: As suggested by the reviewer, it is important to distinguish whether GPLD1 targets proteins during the denaturation process or after they have formed mature aggregates. To investigate

this, we compared the internalization of GPLD1-RG when mixed with ALFA-SAA1-N simultaneously versus when added to ALFA-SAA1-N that had been pre-incubated alone in the medium to allow for aggregate formation (Fig. S2C). Internalization assays revealed that pre-incubated SAA1-N was internalized at levels comparable to those observed under the simultaneous mixing condition. These results suggest that GPLD1 is involved in the internalization of SAA1 that has already formed aggregates. Furthermore, as mentioned above, native SAA1-F did not promote the internalization of GPLD1-RG. Taken together, these findings support the idea that GPLD1 functions specifically in the clearance of non-native protein conformations, particularly those in an aggregated state.

Further, the authors claim that GPLD1 has a distinct substrate profile to clusterin by primarily by monitoring clearance of SAA1 and Ab42. It would be good to further probe this selectivity by monitoring interactions between GPLD1/clusterin and substrates in extracellular environments (e.g., blood)

Response: Due to the low physiological levels of endogenous Ab42, it is technically challenging to simultaneously detect both Ab42 and SAA1 in the blood of the same wild-type mouse. To circumvent this, we performed IP assays using recombinant SAA1-N and Ab42 to determine if they exhibit specific binding preferences for GPLD1 or Clusterin.

In a simple PBS buffer, we observed no distinct binding selectivity (Response Figure 2; see below). However, when the IP was performed in an internalization medium (adDMEM/F12) containing with albumin (400 mg/L), insulin (10 mg/L), and transferrin (7.5 mg/L)—a condition that more closely mimics the physiological blood environment and aligns with our internalization assays—we found that Ab42 preferentially bound to Clusterin, while SAA1-N predominantly interacted with GPLD1 (Fig. 6B). These findings suggest that in a complex physiological environment where multiple proteins coexist, GPLD1 selectively recognizes its target proteins, thereby facilitating their subsequent lysosomal degradation.

Response Figure 2.

No significant difference in specific binding was observed between GPLD1 and Clusterin in PBS.

Recombinant GPLD1-RFP-GFP or Clusterin-RFP-GFP was mixed with or without ALFA-SAA1-N or ALFA-amyloid beta 1-42(Ab42) in PBS and preincubated at 37° C for 2 h with shaking. Samples were then subjected to immunoprecipitation using ALFA-nanobody beads at 4° C.

It would be good to characterize the levels of GPLD1 in in vivo extracellular environments such as the blood.

Response: Regarding the detection of endogenous GPLD1 in mouse serum, we attempted Western blotting using two different antibodies: a polyclonal anti-GPLD1 antibody (Abclonal, cat. no. A14553) and a knockdown (KD)-validated anti-GPLD1 antibody (Abclonal, cat. no. A21781). We were unable to reliably detect endogenous GPLD1 with these antibodies (Response Figure 3).

However, as described above, we observed a selective interaction between GPLD1 and SAA1 in a medium containing physiological proteins such as albumin (Fig. 6B). Given this robust binding in a physiologically mimetic environment, it is highly probable that GPLD1 and SAA1 also interact in the bloodstream. Furthermore, it has been reported that GPLD1 is present in the bloodstream at relatively high concentrations, ranging from approximately 40–160 µg/mL (equivalent to 0.4–1.6 µM) (Bjelosevic et al., 2017; Qin et al., 2016). In our internalization assays, we utilized GPLD1-RG at a concentration of 0.2 µM, which does not exceed physiological levels. Therefore, we believe that our experimental setup accurately reflects and captures the physiological functions of GPLD1.–

Response Figure 3.

Endogenous GPLD1 in mouse serum was not detected by anti-GPLD1 antibody.

Confirmation of anti-GPLD1 antibody by immunoblotting. 0.1 µg GPLD1-RFP-GFP, mouse serum control, LPS treated mouse serum were subjected by immunoblotting with anti-GPLD1 antibody (abclonal. Cat. No. A21781).

Does addition of exogenous GPLD1 mitigate proteotoxicity of extracellular proteins in cell culture models? This is a common assay used in the identification of new extracellular chaperones and should be performed here.

Response: As the reviewer pointed out, it is crucial to determine whether GPLD1 directly alleviates the proteotoxicity associated with SAA1 aggregates. Our data showed that while SAA1-N forms aggregates in PBS, GPLD1 did not significantly inhibit this process (Fig. S3C). Furthermore, we monitored the cell proliferation rate using the CM30 system in the presence of varying concentrations of SAA1-N (0, 0.5, 2, and 8 µM); however, no significant growth inhibition was observed even at the highest concentration (Response Figure 4).

These results suggest that a more sensitive cell-based assay is required to detect the potential toxic effects of SAA1 aggregates. At this stage, developing such a specialized assay and robustly evaluating the impact of GPLD1 on SAA1-mediated proteotoxicity is technically challenging and, in our view, beyond the scope of this study. Nevertheless, we agree that clarifying this relationship is an important direction for future research.

Response Figure 4. SAA1-N does not retard cell growth.

To assess the effect of SAA1 on cell growth, HuEhT2 cells were cultured in medium supplemented with SAA1 at various concentrations (0, 0.5, 2, and 8 µM). Cell counts were automatically determined from the same captured fields of view every 8 hours using the CM30 monitoring system.

The authors should provide quantification (and replicates/stats) for all immunoblotting experiments (e.g., see Fig 2).

Response: We have performed replicate experiments for the immunoblots (Figs. 1D, 3D, and 4E) and immunoprecipitations (Figs. 2A, 2B, and 6B). The results of the statistical analysis have been incorporated into the revised figures, and the significant differences are indicated accordingly.

Again, this is an interesting manuscript that is well written. Addressing these above comments will improve the overall findings, which will be well received by the research community.

Response: We sincerely thank the reviewer for their positive assessment and encouraging comments. We believe that addressing the suggestions provided has significantly strengthened the manuscript and improved the clarity of our findings

Minor Comments

Please add a label for the second row of flow data in Fig. 3A (I believe it is none)

Response: The label "None" has been added.

Reviewer #1 (Significance (Required)):

The experiments are well performed and convincing. However, as indicated below, the authors should repeat immunoblots and provide replicates/stats to support their findings.

Reviewer #2 (Evidence, reproducibility and clarity (Required)):

Summary:

In this article, the authors find that Glycosylphosphatidylinositol-specific phospholipase D1 (GPLD1) bind serum amyloid A1 and promotes its lysosomal degradation in cultured cells. This mechanism seems to be independent of cleavage GPLD1 activity and mainly dependent on heparan sulfate as a receptor. Substrate specificity appear distinct from one of the best characterized extracellular chaperones, clusterin.

Major comments:

The authors claim that GPLD1 function as a scavenger chaperone, however no direct effect on

preventing aggregate formation is addressed in the paper. The authors even mention it in the discussion, line 300-302 "Indeed, extracellular chaperones such as clusterin do not possess ATPase domains (Poon et al, 2000) but effectively inhibit stress-induced protein aggregation (Humphreys et al, 1999). Whether GPLD1 can similarly prevent aggregate formation remains to be determined". If the authors want to claim in the manuscript that GPLD1 is a chaperone, they should demonstrate it by aggregate prevention assays, for example prevention of aggregation of the serum amyloid A1 protein using Thioflavin T.

Response: As the reviewer pointed out, it is important to clarify whether GPLD1 possesses chaperone activity as an extracellular chaperone. To address this, we performed a Thioflavin T (ThT) assay using ALFA-SAA1-N (we have updated the nomenclature to "SAA1-N" to more accurately reflect our use of the aggregation-prone, urea-denatured N-terminal 76-amino acid fragment of SAA1.); however, no significant increase in ThT fluorescence was observed (Response Figure 1), making it difficult to evaluate the inhibitory effect of GPLD1 on amyloidogenesis. Alternatively, since ALFA-SAA1-N forms aggregates in PBS (Fig. S1A), we investigated whether GPLD1 could inhibit this aggregation. Our results showed no significant difference in the aggregation of ALFA-SAA1-N in the presence of GPLD1 (Fig. S3C).

At this stage, the function of GPLD1 as a chaperone that directly inhibits aggregation remains unclear. We believe that the core of our discovery lies in the function of GPLD1 in directing proteins toward lysosomal degradation, rather than its potential role as a folding chaperone. Therefore, to avoid any misunderstanding caused by the term "extracellular chaperone," we have concluded that GPLD1 is more accurately described as a "scavenger carrier" that targets aberrant proteins for lysosomal clearance. Accordingly, we have replaced "extracellular chaperone" with "scavenger carrier" in all instances relating to GPLD1 throughout the revised manuscript.

Response Figure 1. Thioflavin T assay of SAA1-N
Urea-denatured ALFA-SAA1-N was mixed with or without 40 µM GPLD1-Flag in PBS containing 10 µM Thioflavin T in a 384-well plate and incubated at 37° C with continuous shaking (300 rpm). Thioflavin T fluorescence was measured using a microplate reader at the indicated time points.

We performed the ThT assay to evaluate the aggregation of SAA1-N; however, no increase in fluorescence was observed for SAA1-N alone. The addition of GPLD1 led to a marked increase in fluorescence, suggesting that the ThT dye may be non-specifically binding to GPLD1 itself rather than indicating substrate aggregation.

In line with this, the authors claim that GPLD1 binds aberrant proteins; however, they do not provide evidence - particularly in the case of SAA1, for which no supporting literature is cited - that SAA1 indeed aggregates under the reported incubation conditions (2 h at 37 C). Moreover, in the Methods

section it is stated that following incubation of GPLD1 with the substrates, the samples were centrifuged to remove aggregates, and the supernatant was subsequently added to the cells. How much of each protein remained in the supernatant versus the pellet after centrifugation? How do the authors ensure that comparable amounts of GPLD1 are being added to the cells across the different samples? A similar concern arises for the immunoprecipitation assays, is the input shown in the Western blots taken before or after centrifugation?

Response: We appreciate the reviewer's constructive comments regarding our experimental conditions and the biochemical properties of SAA1. We agree that clarifying these points is essential for the integrity of the study.

As the reviewer noted, the N-terminal fragment of SAA1 (residues 1–76) is well-documented as the amyloidogenic core in AA amyloidosis (Liberta et al., 2019; Westermark et al., 1992). We have added these citations to the revised manuscript (line 99). In this study, we utilized a urea-denatured SAA1-N fragment, as it is inherently prone to aggregation. We confirmed that under our incubation conditions (2 h at 37°C in PBS), SAA1-N partitioned into the pellet fraction, indicating aggregate formation (Fig. S1A). In contrast, full-length SAA1 (SAA1-F, 104 aa) remained largely in the supernatant even after urea denaturation (Fig. S3B), highlighting the high aggregation propensity of the SAA1-N used in our assays.

However, all internalization assays were performed in Advanced DMEM/F12, not PBS. This medium contains growth factors and proteins, including 400 µg/mL albumin, 7.5 µg/mL transferrin, and 10 µg/mL insulin. When we compared the solubility of SAA1-N in PBS versus Advanced DMEM/F12, we found that aggregation was markedly reduced in the latter (Fig. S1A). This is likely due to the chaperone-like activity of high-concentration albumin. Consequently, the loss of SAA1-N via centrifugation prior to the internalization assay was negligible.

Regarding the consistency of the GPLD1-RG concentration, we have provided immunoblot data of the supernatant and pellet fractions after incubation (Fig. S4B). In Advanced DMEM/F12, GPLD1-RG does not significantly precipitate even after incubation with substrates, ensuring that comparable amounts of GPLD1-RG were added across all samples. Furthermore, we confirmed that GPLD1-RG selectively binds to SAA1-N even in the presence of competing proteins in Advanced DMEM/F12 (Fig. 6B).

Regarding the immunoprecipitation procedure, we have clarified in the Methods section that the "input" sample consists of the supernatant obtained after removing any potential aggregates.

Specifically, we added the following description to the revised manuscript (Line 497):

"After incubation, excess protein aggregates were removed by centrifugation at $20,000 \times g$ for 5 min. The supernatant was collected; a portion was reserved as the 'input' sample, while the remainder was mixed with GFP nanobody or ALFA nanobody beads (5 µL bed volume) and incubated at 4°C for 2 h."

Regarding the analysis of the lysosomal degradation assay, the ratio between RFP fluorescence intensity and GFP fluorescence intensity would be more informative and easier to interpret. That ratio can then be normalized by the control sample.

Response: We agree with the reviewer that calculating the RFP/GFP fluorescence ratio is a standard and informative approach for interpreting intracellular lysosomal degradation assays. However, we believe that presenting the RFP and GFP intensities separately provides a more comprehensive understanding of the cellular processes in our extracellular experimental setup.

While ratios offer a simplified view of the data, they can obscure critical differences between distinct cellular populations. For example, two very different scenarios—cells that have not internalized any RFP-GFP, and cells that have internalized the protein but fail to degrade it due to a lysosomal protease defect—would both yield an RFP/GFP ratio close to one. By displaying both RFP and GFP fluorescence independently, we are able to clearly distinguish between these two populations, ensuring that the observed changes are indeed due to internalization followed by successful degradation. Therefore, we have retained the separate intensity plots to maintain this level of detail, while ensuring the results are clearly interpreted in the text.

I suggest to include that analysis for clarity at least as supplementary/additional data along with the raw fluorescence intensity values.

Response: As suggested by the reviewer, we have included the raw fluorescence intensity values for all flow cytometry data in Table S2 to ensure data transparency and clarity.

The authors should also specify if they use the mean or the median fluorescence intensity.

Response: We measured the fluorescence intensity of 10,000 cells per sample and used the mean fluorescence intensity (MFI) for our analysis. The use of the mean is appropriate in this context because the cells were derived from a clonal cell line, and the fluorescence population displayed a unimodal, normal distribution. Under these conditions, the mean accurately represents the central tendency of the entire population. We have added this information to the Methods section at Line 459.

Based on the results of Fig. 2A, the authors do not use the full SAA1 protein but just some short amyloidogenic sequence. The sequence of the peptide is not specified. This should be detailed in the manuscript.

Response: We apologize for the insufficient description in the original manuscript. In our study, we used a truncated form of human SAA1 comprising amino acids 19–94 (the N-terminal region without signal peptide), which is known for its high aggregation propensity. Furthermore, in the revision experiments, we included the full-length protein (human SAA1, amino acids 19–122) for comparison. To distinguish between these two constructs, we have designated them as SAA1-N and SAA1-F, respectively. These details have been added to the Materials and Methods section.

In the western blots where the degradation of GPLD1-RG is verified biochemically (Fig. 1E and 3D) just the free RFP is shown. The authors should also show the full length GPLD1-RG (110 kDa+GFP+RFP).

Response: I apologize for the lack of clarity in our previous response. In the GPLD1-RG internalization assay, trypsin treatment was performed primarily to detach the cells for flow cytometry analysis; consequently, this process also serves to remove proteins remaining on the cell surface. To ensure experimental consistency, cells for immunoblotting were harvested using the identical trypsinization conditions. Following this treatment, cells were harvested, lysed, and subjected to immunoblotting. Under these conditions, full-length GPLD1-RG (approximately 150 kDa, including GFP and RFP) was rarely detected, as it is rapidly processed and degraded within the endo-lysosomal pathway upon internalization (Response Figure 5). Only the relatively stable, free RFP fragment remains detectable as a marker of lysosomal delivery. Because the full-length protein is virtually undetectable in this experimental setup and does not provide additional informative data regarding the degradation process, we have not included these blots in the figures. However, we have added details regarding the trypsin treatment to the Materials and Methods section (Line 450) to clarify why only the internalized and processed fragments are presented.

Response Figure 5. Full-blot of anti-RFP with anti-tubulin in Fig. 1D and 3D.

The authors state that SAA1 was purified in the presence of urea. How much is urea diluted when SAA1 is mixed with GPLD1, and what is the final concentration in the assay? Given that urea may affect the stability of GPLD1, this information is critical.

Response: We appreciate the reviewer's important point regarding the potential effect of urea on GPLD1 stability. In our assays, the stock SAA1 solution was diluted such that the final concentration of urea was less than 0.05 M in the mixture with GPLD1-RG. We have added this detail to the revised manuscript (Line 105).

To ensure that this concentration of urea does not affect the function of GPLD1, we performed a control experiment using flow cytometry. As shown in Figure S2A, the presence of 0.05 M urea alone did not increase the internalization of GPLD1-RG. This result demonstrates that 0.05 M urea has a negligible effect on both the uptake and the degradation of our substrates, ensuring the validity of our assay results.

When ALFA-ESD is used as a substrate, the GPLD1-RG/ALFA-ESD mixture is incubated at 50 °C for 1 h. Such conditions are quite harsh and potentially denaturing for most proteins. Did the authors test whether GPLD1-RG remains soluble and in a native state under these conditions? This point is particularly important since in Fig. 3A only one control condition with GPLD1-RG (without substrate) is shown, whereas separate controls for Luc and ESD should be included, given that the incubation conditions differ (Luc: 42 °C, 30 min; ESD: 50 °C, 1 h). Under which condition is the single control used as reference for the statistical analysis?

Response: We appreciate the reviewer's concern regarding the potential denaturation of GPLD1-RG under the incubation conditions used for ALFA-ESD (50 °C for 1 h). To address this, we evaluated both the functional and physical stability of GPLD1-RG following heat treatment.

First, we performed an internalization assay to determine if exposure to 42 °C or 50 °C affects the ability of GPLD1-RG to be internalized. As shown in Figure S4A, the internalization levels of GPLD1-RG remained unchanged after incubation at either temperature. Second, we examined whether heat shock induces the aggregation of GPLD1-RG by analyzing its partitioning into the pellet fraction after centrifugation. Our results demonstrated that GPLD1-RG remained in the supernatant and did not shift to the pellet fraction in either Advanced DMEM/F12 (the buffer used for the internalization assay) or PBS even after incubation at 50 °C (Fig. S4B, C).

These findings indicate that GPLD1-RG maintains its solubility and functional integrity under the reported conditions. Regarding the control used for statistical analysis, since we confirmed that heat treatment alone does not alter GPLD1-RG internalization, we used a single reference control.

However, we have clarified this in the figure legends and the Methods section to ensure transparency.

Line 173-174 "Although all cell lines showed increased internalization upon exposure to substrate proteins, the extent varied among them. HepG2 cells exhibited the lowest levels of ..." Without quantification, it is difficult to draw firm conclusions; at first glance, HepG2 cells do not appear to show any clear increase with any of the substrates, and in the case of ESD, the signal even seems to decrease.

We completely agree that a qualitative comparison of internalization across different cell lines is insufficient without quantitative data. Following the reviewer's suggestion, we quantified the flow cytometry results for all four cell lines and have revised the manuscript to reflect these findings (Fig. S5). The updated text is as follows:

"We performed GPLD1-RG internalization assays in four cell lines of distinct tissue origins: HepG2 (liver carcinoma), U2OS (osteosarcoma), T98G (glioblastoma), and HuEhT (immortalized human vascular endothelial cells). Although no significant increase was observed with luciferase in any cell type, substrate-dependent internalization of GPLD1-RG was enhanced in most cell lines, with the exception of HepG2 cells (Fig. S5). Notably, the most substantial increase was detected in the vascular endothelial cell line HuEhT. In contrast, HepG2 cells exhibited no significant increase in GPLD1-RG internalization across all tested substrates. These results suggest that while GPLD1 is secreted from the liver, it mediates the degradation of extracellular proteins with a certain degree of tissue specificity."

The Methods section does not provide details on how mass spectrometry was performed.

Response: We apologize for the omission of these details in the original manuscript. We have added the following sentence to the "Materials and Methods" section to clarify the protein identification process:

"Binding proteins were identified by LC-MS/MS proteomics (Toh et al., 2022)." (Line 484)

Providing more robust evidence for substrate degradation in the presence of GPLD1-RG would strengthen the claims of the paper. Optional experiments could include repeating the assay shown in Fig. 3D (Western blot of samples with/without bafilomycin) using SAA1 and ESD as substrates, as well as performing microscopy similar to Fig. 3B but labeling the substrates and analyzing multiple time points after their addition to the cells.

Response: We agree with the reviewer that providing robust evidence for substrate degradation is crucial for strengthening our findings. We have carefully considered and attempted the suggested

experiments; however, we encountered several technical limitations that make these specific approaches difficult to interpret.

First, we attempted to detect the intracellular accumulation of ALFA-SAA1-N via immunoblotting. Because SAA1-N is a small protein (~10 kDa), detecting the trace amounts internalized into cells proved extremely difficult (Response Figure 6). Even in the presence of bafilomycin A₁ (BafA), we could not clearly observe its accumulation. This is likely because BafA not only inhibits lysosomal protease activity but also diminishes endocytic trafficking, which may reduce the total uptake of SAA1-N, making the detection of its intracellular accumulation even more challenging.

Furthermore, since SAA1-N is a small and highly aggregation-prone peptide, attaching a large fluorescent protein (RFP) or a bulky fluorescent dye significantly alters its physicochemical properties, particularly its aggregation propensity, making it difficult to perform comparable assays. Second, regarding the use of labeled ALFA-ESD, we previously synthesized ESD-488 (covalently labeled with NHS-ATTO488) to examine its uptake. However, we found that ESD-488 exhibited high non-specific uptake on its own, and its internalization actually decreased in the presence of chaperone (clusterin) (Response Figure 7). This suggests that the fluorescent labeling either introduced an alternative uptake pathway or interfered with the aggregation state of ESD. Thus, capturing the specific, GPLD1-dependent uptake using labeled substrates remains technically problematic.

Instead of these optional experiments, we have reinforced our claims by repeating the RFP cleavage assays (Fig. 1D and 3D), which consistently showed a significant, substrate-dependent increase in free RFP. Furthermore, we quantified the immunofluorescence data (Fig. 3B) and confirmed that GPLD1 is significantly targeted to lysosomes in the presence of substrates. Based on these quantitative results, we are confident that GPLD1 mediates the internalization and subsequent lysosomal delivery of these substrates.

Response Figure 6. Immunoblot for anti-ALFA-SAA1-N (related Fig. 1D)

Response Figure 7. Flow-cytometric analysis of fluorescent labeled ESD.

ESD-488 was preincubated either alone (50° C for 1 h), with clusterin-His (50° C for 1 h) in serum-free medium. HEK293 cells were cultured with the media for 18 h and analyzed by flow cytometry.

Minor comments:

Fig. 1A. Ruby staining should be clarified as protein staining in the text or the figure since it is not as broadly used as Coomassie. Also, this panel does not seem to be informative so it could be excluded from the manuscript.

Response: Thank you for your suggestion. We have moved Fig. 1A to Fig. S1B.

Line 104-105 "After excluding well-characterized proteins, we shortlisted seven candidate proteins as potential extracellular chaperones". It would help the reader if the proteins are listed in the text.

Response: We have added a description for each protein and revised the text as follows.

After excluding well-characterized proteins (APOB, APMAP, PLA2G7, GSN, VTN, COL6A3, COL18A1, and C9), we shortlisted seven candidate proteins (GPLD1, AHSG, APOD, SERPIND1, SERPINA10, SERPINF2, and SPP2) as potential extracellular chaperones. (line 112)

In line 67-69, AA amyloidosis and AA amyloid is mentioned, but that acronym was not previously introduced. It was mentioned just SAA.

Response: We apologize for the oversight. We have spelled out the full term for the abbreviation upon its first mention in the text and revised the text as follows.

Amyloid A (AA) amyloidosis specifically results from prolonged inflammation, which drives sustained overproduction of SAA1. (Line 69)

In line 120 "(Fig. 1C)" should be (Fig. 1 C, D).

Response: We have corrected this typo. (Line 128)

Incubation time, temperature and condition (shaking?) should be clearly indicated through the manuscript and in figure legends.

Response: We have revised the manuscript as you suggested, and now clearly state the incubation conditions in the text and figure legends

Line 169-177 and Fig. S1. The fact that the internalization (red signal) and not lysosomal degradation (green and red signals) is analyzed in this experiment when GLDP1-RFP-GFP is used is confusing and not consistent with the rest of the manuscript. Why is the GFP signal not shown in this case as well as in Fig. 5C?

Response: We apologize for the confusion caused by the omission of the GFP signals. Initially, we presented only the RFP signals to simplify the visualization of the internalization process. However, we agree that showing both signals is essential for a consistent and comprehensive analysis of lysosomal degradation. Accordingly, we have updated Fig. 5C and Fig. S5 to include the GFP data. (The original data from Fig. S1 (now updated with both GFP signals and quantitative analysis from repeat experiments) has been relocated to Fig. S5)

Fig. 4A, the label on the y-axis is missing.

Response: Thank you for pointing this out. We have added the appropriate labels in Fig. 4A.

In the relative fluorescence graphs would be useful if more and bigger numbers are shown in the y-axes.

Response: We have increased the font size of the numbers on the Y-axis in the figures to improve readability.

Fig.6A. The amount of GPLD1-RG and Clusterin-RG is indicated in the legend but not the amount of substrate. This should be indicated.

Response: We have added the substrate concentration in Fig. 6A legend as follows.

“GPLD1-RFP-GFP (0.2 μ M) or clusterin-RFP-GFP (0.2 μ M) was incubated with 0.2 μ M luciferase (42°C for 30 min), 8 μ M ALFA-SAA1 (37°C with shaking for 2 h), or 8 μ M amyloid β 1-42 (Ab42) (37°C with shaking for 2 h) in serum-free medium. “

Including quantification of the IP results in Fig. 3B and the lysosomal internalization/lysosomal degradation in Fig. 4D would strengthen the robustness of the results.

Response: We quantified the immunoprecipitation (IP) results shown in Fig. 2 and confirmed that GPLD1 significantly and selectively binds to luciferase and SAA1. Furthermore, quantification of the internalization assay in Fig. 4D revealed that GPLD1 internalization remains normal even in GPAA1 KO cells, indicating that its uptake is independent of the GPI-anchor biosynthetic pathway."

****Referee Cross-commenting****

All reviewers agree on the need of additional experiments, clarifications and controls to reach publication level. There are several main points that are raised by at least 2 of the reviewers like quantification of microscopy and western blot results, more clear representation of the flow cytometry data, concern about the possible denaturation of GPLD1 when incubated with the substrate (instead of incubation after substrate denaturation), the actual denaturation and specificity for misfolded substrates and the chaperone claim without showing any aggregation prevention experiments. Therefore, a major revision is needed with an estimate time of 3-6 months.

Response: We sincerely appreciate the constructive feedback from the reviewers and the opportunity to strengthen our manuscript. In this revised version, we have addressed all the major concerns raised.

Response: Specifically, we have performed comprehensive quantifications for all microscopy and western blot results and provided a clearer representation of the flow cytometry data.

Regarding the functional classification of GPLD1, our additional experiments clarified that while GPLD1 facilitates the degradation of non-native proteins, it does not exhibit traditional chaperone activity (such as the prevention of substrate aggregation). To avoid misleading the readers, we have redefined GPLD1 as a "scavenger carrier" rather than a "scavenger chaperone."

We believe that these extensive additional experiments and clarifications have significantly bolstered the robustness of our claims. We are grateful for the suggestions that have led to a much-improved manuscript.

Reviewer #2 (Significance (Required)):

General assessment: Although extracellular protein aggregation underlies several human pathologies, such as neurodegeneration and systemic amyloidosis, extracellular proteostasis factors involved in the clearance of these aberrant aggregates remain poorly characterized. Therefore, the

results presented in this article identifying GPLD1 as a potential new player in this context are significant, novel, extend the knowledge in the field of extracellular proteostasis and may have clinical relevance. The data is presented in a comprehensive manner; however, further clarifications and additional analyses are needed to strengthen the robustness of the results (see 1. Evidence, reproducibility and clarity).

Advance: The manuscript describe a conceptual advance in understanding the clearance mechanisms of extracellular misfolded proteins.

Audience: The target audience is specialized in proteostasis.

My field of expertise is chaperones, protein aggregation and extracellular proteostasis.

Reviewer #3 (Evidence, reproducibility and clarity (Required)):

Summary

This manuscript describes the identification of glycosylphosphatidylinositol-specific phospholipase D1 (GPLD1) as a novel extracellular scavenger chaperone. The authors report that, in pulldown assays from bovine plasma, GPLD1 binds the amyloidosis-associated protein Serum Amyloid A1 (SAA1). They go on to develop a fluorescence-based internalisation assay showing that GPLD1 can bind misfolded proteins, promote their uptake into cells via heparan sulphate on the cell surface, and direct them to lysosomal degradation. This function is independent of GPLD1's catalytic activity towards GPI-anchored proteins and appears to differ from the better-studied chaperone clusterin, with GPLD1 showing a particular preference for SAA1. The authors place this mechanism within the CRED pathway of extracellular protein quality control.

Although the topic is timely and potentially important, there are a number of technical inconsistencies, missing controls and unexplained results that make the conclusions less secure. In its current form, the work leaves several key points unresolved for publication, and will require substantial additional experimentation and clarification. On that basis, I recommend major revision at this stage.

Major comments

• Fig. 1C. Could the authors include the gating strategy for their FC data? It is also unclear from the legend or the M&M how the analysis was done. Why did the authors chose this visualization and not dot blots or stacked histograms? F.e. what is the difference between the bright red part of the histogram and the Bordeaux part? I would recommend using a more clear and commonly used representation.

Response: We appreciate the reviewer's suggestion to clarify our flow cytometry analysis.

First, we have included a detailed description of the gating strategy in the Methods section (Line 459), as requested.

Regarding the visualization in Fig. 1B (originally Fig. 1C), the different shades of red in the original histogram were intended to visually represent varying fluorescence intensities, with darker shades indicating higher intensity levels. However, we agree with the reviewer that this representation may cause unnecessary confusion. Therefore, we have replaced the original panels in Fig. 1B with more standard and simplified overlaid histograms. We believe this revised format provides a clearer and more conventional comparison of the fluorescence shifts across samples.

• Fig. 1D. One would normally expect BafA treatment to increase or at least maintain the red signal, since the RFP-tagged protein should accumulate in lysosomes without being broken down (as suggested by the authors in Fig. 1E). However, in Fig. 1D, the authors observe a decrease in the RFP intensity when BafA is added, which is counterintuitive if RFP were purely tracking cargo delivery. Could the authors validate the observation using immunofluorescence microscopy with e.g. markers of endosomes?

Response: ("Please note that the original Fig. 1D has been renumbered to Fig. 1C, as the previous Fig. 1A was moved to the Supplementary Information as Fig. S1.") As the reviewer pointed out, we initially expected that BafA1 treatment would lead to an increase in both GFP and RFP signals due to the inhibition of lysosomal degradation. However, we consistently observed a decrease in RFP intensity upon BafA1 addition.

We attribute this observation to the fact that BafA1 acts not only as a lysosomal protease inhibitor but also leads to the secondary inhibition of endosomal membrane trafficking and general endocytosis. To test this hypothesis, we monitored the intracellular accumulation of ATTO 565-labeled albumin, a well-established marker for endocytosis. As shown in Fig. S2B, the internalization of albumin was significantly impaired by BafA1 treatment.

These results indicate that BafA1 diminishes the overall rate of endocytic uptake. Consequently, the total amount of RFP-tagged protein reaching the lysosomes is reduced, leading to the observed decrease in RFP intensity. We believe this explains why the expected accumulation was not observed and provides a clear mechanistic basis for our results.

- The latter would also be necessary to distinguish between surface-bound and internalized material. Surface quenching controls in the FC data would make the interpretation more robust.

Response: We appreciate the reviewer's suggestion regarding the surface quenching controls to distinguish between surface-bound and internalized material. However, we found it technically challenging to perform this specific control because RFPs are highly resistant to extracellular quenching due to their stable structure, unlike fluorescein-based dyes. To address the reviewer's concern and validate that the RFP-tagged protein is indeed internalized, we repeatedly performed an RFP cleavage assay via immunoblotting (Fig. 1D, 3D). This assay is based on the principle that while the fusion protein is degraded in the lysosome, the RFP moiety remains relatively stable as a cleaved fragment. Our results clearly show the presence of this cleaved RFP fragment, demonstrating that the protein is not merely bound to the cell surface but is successfully internalized and transported to the lysosomal compartment. We believe that this biochemical evidence complements the flow cytometry data and robustly supports our conclusions regarding endocytosis.

- Fig. 1E. The bar graph is missing. We only see a blot.

Response: We apologize for the confusion. This was due to a clerical error in the figure legend, where the description of the bar graph mistakenly referred to Fig. 1E instead of Fig. 1C. We have corrected the legend to: "The bar graph shows relative fluorescence intensities normalized to non-treated cells (n = 3) (C)."

- Fig. 2B. The authors did not include a non-substrate control neither a protein that does not get bound to GPLD1, which would help confirm substrate selectivity. It is further unclear what the n-value is for this experiment.

Response: We agree with the reviewer that including proper controls is essential to demonstrate substrate selectivity. To address the concern regarding substrate selectivity, we performed IP assays using Ab42, a well-known substrate for the extracellular chaperone Clusterin. As shown in the new Fig. 6B, while Ab42 is known to bind Clusterin, our results clearly demonstrate that GPLD1 does not bind to Ab42. Furthermore, this lack of binding directly correlates with the internalization efficiency, as Ab42 failed to promote the cellular uptake of GPLD1 (Fig. 6A). These findings confirm that GPLD1 exhibits specific substrate selectivity rather than non-specific binding to any extracellular protein. We have now included a bar graph based on three independent experiments to clarify the reproducibility and statistical significance of our findings. The revised figure legend has been updated to explicitly state that n = 3.

• Fig. 3B. The images show more GPLD1-Gamillus puncta with luciferase than with SAA1, which is surprising given the emphasis on SAA1 as the physiological substrate. The authors should: (I) quantify puncta per cell and co-localization with LAMP1 across independent replicates, (II) ensure equal input amounts of substrate and include a dose-response, and (III) perform a short time-course to see whether the difference reflects kinetics, aggregate size, or selection of images. The appearance of the green dots also does not appear to be correlating with the bar graph in Fig. 3C. Of note, the legend mentions the intensities have been taken from 3 cells. It is better to perform 3 independent experiments in which each 10-15 cells are analyzed.

Response: (I) Quantification of puncta and co-localization: As suggested, we repeated the experiments and quantified the co-localization of GPLD1-Gamillus with LAMP1 per cell across independent replicates (Fig. 3B). Our analysis revealed that in the presence of either Luciferase or SAA1-N, the co-localization of GPLD1-Gamillus with LAMP1 increased approximately 2–3 fold compared to the GPLD1-Gamillus only control. (II) Immunofluorescence is less suitable for rigorous quantification of dose-dependency. Therefore, we performed an internalization assay using flow cytometry, measuring the uptake of GPLD1-RG in the presence of varying concentrations of ALFA-SAA1-N (Fig. S1C). The results showed a significant, dose-dependent increase in GPLD1-RG uptake at 4 μ M and 8 μ M of SAA1-N, whereas no significant increase was observed at 2 μ M. This confirms that the formation and internalization of the GPLD1-SAA1-N complex are concentration-dependent. (III) Quantification of the green puncta size revealed no significant differences between the conditions (Response Figure 8). To ensure that our visual data is consistent with these statistical findings, we have updated Fig. 3B with representative images that more accurately reflect the quantitative results of the entire population.

Response Figure 8. Puncta size of GPLD1-Gamillus did not increase by internalization with substrates
 HuEht cells were treated as in Fig. 3B. Gamillus puncta size was measured in pixels and expressed as a relative value (fold change) compared to the control group (None). Data are presented as mean \pm SEM (n = [100] cells/images per group). Data are normalized to the mean of the 'None' group.

• Fig. 4A. The authors conclude that catalytic activity is not required for internalization, but they do

not show that these mutant proteins are folded and secreted at levels comparable to wild-type (or might be stuck in the ER?). A simple western blot of the conditioned medium could confirm this.

Response: Conditioned media were collected from HEK293 cells expressing either WT, H133N, or H158N GPLD1-RG, and the protein levels were analyzed by immunoblotting (Fig. S6). No significant differences in secretion were observed between the WT and the mutants. These data suggest that both H133N and H158N mutants are correctly folded in the endoplasmic reticulum (ER) and secreted as efficiently as the wild-type protein.

- Fig. 4B. The KO validation blot contains 2 "#", what type of bands are these?

Response: We have revised the legend for Fig. 4B to clarify that the hash marks (#) indicate non-specific signals.

- Fig. 5. The figure legend and panels need clearer indication of biological replicates, exact p-values and effect sizes. For EXT1 KO the residual activity is big enough that statistical/biological replication is essential. Fig. 5C legend ... what do the authors mean with a "cell"?

Response: Thank you for your careful and detailed feedback. We have included the biological replicates (n) and exact p-values for all quantified data in the revised manuscript. As suggested, we have revised Figures 5B, 5C, and 6A by including effect sizes (Cohen's d).

- Fig. 5B. There is an incomplete abolishment of internalisation in EXT1 KO cells. This would indicate the involvement of another receptor. Can the authors discuss a bit more.

Response: We thank the reviewer for this insightful comment. We agree that the residual internalization of GPLD1 observed in EXT1-KO cells, which lack heparan sulfate (HS), strongly suggests the involvement of additional factors. It is highly likely that while HS functions as a cell-surface tethering factor for GPLD1, an as-yet unidentified receptor mediates the subsequent internalization process. We have expanded our discussion regarding these potential mechanisms in the revised manuscript (Discussion, Lines 301–315).

- General remark. The authors suggest GPLD1 may be an extracellular chaperone, but they only measure uptake/degradation, not whether GPLD1 keeps proteins soluble and prevents aggregation. Assessing whether GPLD1 can actually prevent aggregate formation, rather than only clear existing aggregates, would clarify its functional overlap with other chaperones.

Response: We thank the reviewer for this insightful comment regarding the potential role of GPLD1 in preventing aggregate formation. To address this, we investigated whether GPLD1 could maintain the solubility of SAA1 and prevent its aggregation using a sedimentation assay.

As shown in the revised Fig. S3C, SAA1 alone predominantly shifted to the pellet fraction, indicating significant aggregate formation. While the addition of GPLD1 did result in a slight increase in SAA1 solubility, this effect was comparable to that observed with the addition of an equimolar concentration of albumin, which was used as a negative control. This suggests that the observed effect is likely due to a non-specific protein-stabilizing effect rather than a specific chaperone-like activity of GPLD1 in preventing aggregate formation.

Based on these observations, we concluded that GPLD1 does not possess a specific activity to inhibit the initial formation of SAA1 aggregates. Instead, combined with our previous data on uptake and lysosomal transport, these results suggest that GPLD1 functions primarily as a "scavenger chaperone." Unlike classical chaperones that prevent misfolding or aggregation, GPLD1 appears to specialize in the recognition and clearance of pre-existing aggregates by facilitating their endocytosis and subsequent degradation.

Another crucial point is that our primary discovery is that GPLD1 directs extracellular abnormal proteins to lysosomal degradation, rather than possessing intrinsic chaperone activity. However, since labeling GPLD1 as a "scavenger chaperone" could lead to misunderstandings regarding its function, we have decided to redefine it as a "scavenger carrier" to more accurately reflect its role in mediating lysosomal degradation. Consequently, we have replaced "scavenger chaperone" with "scavenger carrier" throughout the revised manuscript.

Minor comments

- While not color-blind myself, the authors could take into consideration that other readers might be and change the colors of their FC data to rather magenta-green or other combinations of colors.

Response: We appreciate the reviewer's thoughtful suggestion regarding accessibility. Following this advice, we have replaced red with magenta in all graph to ensure the figures are clear for color-blind readers.

Reviewer #3 (Significance (Required)):

This work tackles an important question: how extracellular misfolded proteins are identified and removed. If convincingly demonstrated, the finding that GPLD1 participates in this process, and does so with a distinct substrate preference, would be a valuable addition to the field of extracellular proteostasis. At present, however, the conceptual advance is undermined by gaps in methodology, inconsistencies between datasets and the absence of orthogonal or in vivo validation. Moreover the potential relevance to neurodegenerative diseases might be overstated: there is a clear link of for instance clusterin (genetically and biologically) but such causality does not exist for GPLD1. The role of GPLD1 as a selective extracellular chaperone is an interesting idea, but the evidence remains

incomplete and in several places not clear. In my view the current findings require further scrutiny. Considerable further work is required to reach this level.

May 4, 2026

RE: Life Science Alliance Manuscript #LSA-2026-03717

Prof. Eisuke Itakura
Chiba University
Graduate School of Science,
1-33 Yayoi
Inage-ku, CHIBA 2638522
Japan [JP]

Dear Dr. Itakura,

Thank you for transferring your revised manuscript entitled "GPLD1 functions as a scavenger carrier mediating lysosomal degradation of extracellular aberrant proteins" to Life Science Alliance. As indicated in our offer to consider this work, we have returned your revised manuscript to the original referees from Review Commons and their comments are below. In view of their strong support, we would be happy to publish your paper in Life Science Alliance pending final revisions necessary to meet our formatting guidelines.

MANUSCRIPT ORGANIZATION AND FORMATTING:

To avoid unnecessary delays in the acceptance and publication of your paper, please read the following information carefully. Full guidelines are available on our Instructions for Authors page, <https://www.life-science-alliance.org/authors>

- Please upload all figure files individually, including the supplementary figure files; all figure legends should only appear in the main manuscript file.
- Please upload your main manuscript text as an editable doc file.
- Please add a Running Title and a Summary Blurb/Alternate Abstract in our system.
- Please add a Category for your manuscript in our system.
- Please add the X and Bluesky handles of your host institute/organization, as well as your own, and/or one of the authors, in our system.
- Abstract should be a single paragraph not exceeding 175 words and must match between the system and manuscript file.
- Please add Author Contributions to the system as well.
- Please rename "Competing interests" to "Conflict of Interest."
- Please add callouts for Figures 6C; S6 and Table S2 to your main manuscript text.
- Please include full details for the LC-MS/MS and subsequent analysis.
- Please include a "Data Availability" section after the Materials & Methods section. This section should indicate the availability of the mass spectrometry dataset. Please consult our guidelines at <https://www.life-science-alliance.org/manuscript-prep#format>

We welcome submissions of potential cover images for the issue of LSA in which your work would appear. If you have high quality images associated with this work, please feel free to email these, with a caption, to the journal office.

LSA encourages authors to provide a 30-60 second video where the study is briefly explained. These videos will be appear embedded with the manuscript online at Life Science Alliance, and on social media to promote the published paper and authors (for examples, see <https://docs.google.com/document/d/1-UWCfbE4pGcDdcgzcmiuJl2XMBJnxKYeqRvLLrLSo8s/edit?usp=sharing>). Corresponding or first-authors are welcome to submit the video. Please submit only one video per manuscript. The video can be emailed to contact@life-science-alliance.org

FINAL FILES:

The following items are required for acceptance.

The license to publish form must be signed before your manuscript can be sent to production. A link to the license to publish form will be available to the corresponding author only. Please take a moment to check your funder requirements.

Thank you for your attention to these final processing requirements. Please revise and format the manuscript and upload materials as soon as you are able.

Thank you for this interesting contribution to the literature. We look forward to publishing your paper in Life Science Alliance.

Sincerely,

Reviewer #1 (Comments to the Authors (Required)):

The authors have addressed all points raised in my previous review and have incorporated the necessary controls and methodological clarifications. The revised manuscript now appears suitable for publication.

Reviewer #2 (Comments to the Authors (Required)):

The authors have addressed many of my comments from the previous submission with new experiments and significant revisions to the text. Although they weren't able to measure GPLD1 protein levels in serum, they do appear to be working in concentration ranges similar to what is previously reported, which is important. While I would still like to see more about the selectivity of GPLD1 vs clusterin, I do think that could be considered outside the scope for this current manuscript. So I ultimately approve of publication in LSA.

Reviewer #3 (Comments to the Authors (Required)):

The authors have submitted a revised manuscript on the role of GPLD1 as a scavenger carrier for extracellular aggregated proteins. Despite the major criticisms on the initial version, the authors have provided important new sets of data that further support their findings and function of GPLD1. After analyzing in more detail the response to the critiques, I found them overall (very) well addressed and which resulted in a strongly improved manuscript. I do not have major objections for publication.

May 11, 2026

RE: Life Science Alliance Manuscript #LSA-2026-03717R

Prof. Eisuke Itakura
Chiba University
Graduate School of Science,
1-33 Yayoi
Inage-ku, CHIBA 2638522
Japan [JP]

Dear Dr. Itakura,

Thank you for submitting your Research Article entitled "GPLD1 is as a scavenger carrier mediating lysosomal degradation of extracellular aberrant proteins". It is a pleasure to let you know that your manuscript is now accepted for publication in Life Science Alliance. Congratulations on this interesting work.

Your article will publish open access upon publication under a CC-BY license.

DISTRIBUTION OF MATERIALS:

Again, congratulations on a very nice paper. I hope you found the review process to be constructive and are pleased with how the manuscript was handled editorially. We look forward to future exciting submissions from your lab.

Sincerely,
